# Graph-based autoencoder integrates spatial transcriptomics with chromatin images and identifies joint biomarkers for Alzheimer's disease

Xinyi Zhang [1,2], Xiao Wang[1,2], G. V. Shivashankar[3,4] & Caroline Uhler [1,2] ✉

Tissue development and disease lead to changes in cellular organization, nuclear morphology, and gene expression, which can be jointly measured by spatial transcriptomic technologies. However, methods for jointly analyzing the different spatial data modalities in 3D are still lacking. We present a computational framework to integrate Spatial Transcriptomic data using over-parameterized graph-based Autoencoders with Chromatin Imaging data (STACI) to identify molecular and functional alterations in tissues. STACI incorporates multiple modalities in a single representation for downstream tasks, enables the prediction of spatial transcriptomic data from nuclear images in unseen tissue sections, and provides built-in batch correction of gene expression and tissue morphology through over-parameterization. We apply STACI to analyze the spatio-temporal progression of Alzheimer's disease and identify the associated nuclear morphometric and coupled gene expression features. Collectively, we demonstrate the importance of characterizing disease progression by integrating multiple data modalities and its potential for the discovery of disease biomarkers.

Developmental and disease processes are accompanied by changes in the spatial organization and interactions of different cell types and states. These changes are reflected in both a cell's gene expression profile as well as its spatial location within a tissue[1,2]. The recent development of spatial transcriptomic technologies has the potential to provide unprecedented insights into biological processes in tissues at cellular scale. In the context of Alzheimer's disease (AD), early studies, using candidate marker genes in single-molecule FISH experiments, revealed the importance of spatial resolution by identifying the proximity of disease-associated microglia to amyloid plaques[1]. The advent of spatial transcriptomics makes jointly measuring the expression of a large number of genes at high spatial resolution possible. This ranges from several thousand genes with sub-micron resolution in STARmap, seqFISH, and MERFISH to whole-transcriptome

coverage with 10–55 μm resolution in Slide-seq and 10x Genomics Visium[3–8]. The resulting datasets provide an opportunity to identify complex spatial motifs of cells in tissues well beyond the localization of a single cell type. To fully exploit the potential of such spatial transcriptomic data, novel computational methods are required that can incorporate the expression of all genes together with the 3D context.

The analysis of spatial transcriptomic data has so far mainly relied on methods developed for single-cell RNA-sequencing (scRNA-seq) to perform standard workflows such as dimensionality reduction and the clustering of cells/beads/spots for the identification of cell types and states[3,5]. Since such methods take gene expression as input without any spatial context, the use of single-cell methods to analyze spatial transcriptomic data is akin to performing image analysis based on dissociated pixels. Recent studies that allow for the incorporation of

[1]Massachusetts Institute of Technology, Cambridge, USA. [2]Broad Institute of MIT and Harvard, Cambridge, USA. [3]ETH Zurich, Zurich, Switzerland. [4]Paul Scherrer Institute, Villigen, Switzerland. ✉e-mail: cuhler@mit.edu

spatial information perform statistical tests on individual genes to identify genes that are non-randomly distributed in space[9–13]. These methods analyze single genes and do not explicitly learn the composition of cell types and cell states in different tissue neighborhoods. More importantly, the distribution of individual genes can provide only limited insights into tissue region-specific behaviors, for example, with respect to disease progression or response to external stimuli. A different line of computational approaches for spatial transcriptomics aims to infer the types of cells contained in each measured bead/spot for platforms with lower than single-cell resolution[14–16], which is different from our goal of integrating gene expression and cell location for downstream analysis. Recently, autoencoders, prominent neural network architectures that are widely used for representation learning[17], have shown promising results in the context of single-cell scRNA-seq analysis[18–20]. We overcome the limitations of previous methods for spatial transcriptomics by proposing a graph-based autoencoder framework that learns a joint representation of both the expression of all measured genes and the spatial location of cells, such that the separation of cells into different clusters depends on patterns in the expression of combinations of measured genes as well as cellular neighborhoods.

While correcting for batch effects is a critical and standard step in the analysis of scRNA-seq data[19–24], sample-to-sample variations are even more pronounced in spatial transcriptomics given the complexity of sample preparation. Deep learning models for scRNA-seq analysis either require separate pre-processing or use additional modeling parameters to remove batch effects in gene expression[19,25,26]. In addition to the biological and technical variations that are also found in scRNA-seq data, spatial transcriptomics data contains sample-to-sample variations due to differences in tissue morphology, the tissue slicing region, as well as due to distortion, rotation, translation, and/or rupture of the tissue during experimental handling. Correcting for these additional morphological differences between samples is necessary for consistent downstream data analysis and cannot be achieved by the previous batch correction methods for scRNA-seq data. We introduce a method for integrating different samples by overparameterizing a given neural network, i.e. expanding the hidden layer sizes of the network to be larger than the input feature dimension. While intuitively the use of such networks may result in overfitting, they have recently been shown to generalize well and self-regularize[27,28]. Such an approach based on over-parameterization is generally and directly applicable to any neural network architecture. We demonstrate batch effect removal using an over-parameterized graph-based autoencoder for the analysis of STARmap PLUS data taken from different mouse brains.

A relatively unexplored area that is ideally suited for exploration based on spatial transcriptomics data is the coupling between a cell's gene expression and its mechanical microenvironment. For example, nuclear shape and chromatin packing as measured by chromatin staining contains important information about the mechanical microenvironment of a cell and is tightly coupled to its gene expression[29]. Technologies such as STARmap and 10x Genomics Visium measure chromatin staining paired with spatial transcriptomics, but current analysis methods do not make use of all three modalities together (chromatin staining, transcriptomics, and spatial coordinates) or only use images as a cell similarity metric for denoising the gene expression data[30]. Given the coupling between chromatin organization and gene expression, incorporating the chromatin imaging data is critical to fully exploit spatial transcriptomic data and also provides an avenue to study the coupling between the mechanical microenvironment of a cell and its gene expression. We build on our earlier work on combining scRNA-seq and imaging[31] to incorporate the paired chromatin images into our graph-based autoencoder representation of spatial transcriptomic data. A schematic of our approach, which we call STACI (Spatial Transcritpomics combined using Autoencoders with Chromatin Imaging) is shown in Fig. 1b. Once STACI is trained with such multimodal data, the joint representation of gene expression, cell location, and DNA staining can be used to infer missing modalities for new samples, thereby providing an avenue for reducing experimental costs by allowing to perform only chromatin imaging on some samples and inferring the corresponding transcriptomic profiles (Fig. 1c). Importantly, the joint representation provides a powerful approach for the identification of disease biomarkers to track disease progression in different tissue regions using features that combine chromatin packing and gene expression (Fig. 1d).

With the rise of spatial transcriptomic data, several computational approaches have been developed to integrate different data modalities in the tissue context. STACI is the first method, to our knowledge, that simultaneously integrates all the available modalities, namely gene expression, cellular neighborhoods, and chromatin imaging, and is capable of translating between different data modalities and identifying combined morphometric and molecular disease biomarkers in the tissue context. In particular, various methods have been developed that integrate or can be adapted to integrate single-cell gene expression with images through a joint latent space, but do not incorporate cell location into their analysis[32–34]. In addition, other methods such as HMRF[35] incorporate spatial information into the analysis of gene expression to identifying spatial regions with consistent patterns of cell states but do not make use of imaging data. It is the joint latent representation of gene expression and cell location used by STACI that enables us to incorporate chromatin imaging data into the analysis and perform various downstream analysis, such as clustering cells into finer spatial regions, without retraining the model. An interesting method that integrates all three modalities available in spatial transcriptomic datasets is stLearn, a denoising approach that replaces the gene expression of a cell by the average expression of its neighboring cells, weighted by their image similarity[30]. In contrast, STACI aims to identify patterns in cell neighborhoods, which can consist of diverse cell types/states, by taking into account all cells in the physical neighborhood regardless of morphological similarity. Both stLearn and HMRF are unable to predict gene expression from images or identify morphological disease markers associated with the identified tissue regions. Finally, another key feature of STACI compared to current methods for multimodal integration is the built-in correction of batch effects that applies to all modalities in the joint latent space.

As a concrete application, we apply STACI in the context of AD to study its spatio-temporal progression based on a recently released STARmap PLUS dataset taken from TauPS2APP transgenic mice, a model system for AD (Fig. 1a). The STARmap PLUS dataset consists of four mouse brain samples taken from AD and control mice at 8 months and 13 months each (Fig. 2a)[4]. Spatial transcriptomics, chromatin images, the deposition of amyloid-β plaques, and neurofibrillary tau tangles, hallmarks of AD[36,37], were jointly measured in each tissue section. Previous studies observed chromatin condensation, which is associated with apoptosis, in cortical neurons treated with plaques in vitro[38,39]. In addition, disease-specific subtypes of glial cells have been identified by RNA-seq and spatial transcriptomics[1,4,40,41]. These analyses investigated one modality at a time and lack a comprehensive integration of high-throughput sequencing with chromatin imaging in the tissue context. We apply STACI to analyze plaque deposition jointly with gene expression, cell location, and chromatin images in mouse brains, thereby identifying biomarkers of disease progression. Our approach could be applied in the same way to tau tangles or any other protein of interest. In addition, while we here apply STACI to STARmap data, our method is broadly applicable to spatial transcriptomic technologies including Visium and MERFISH[5].

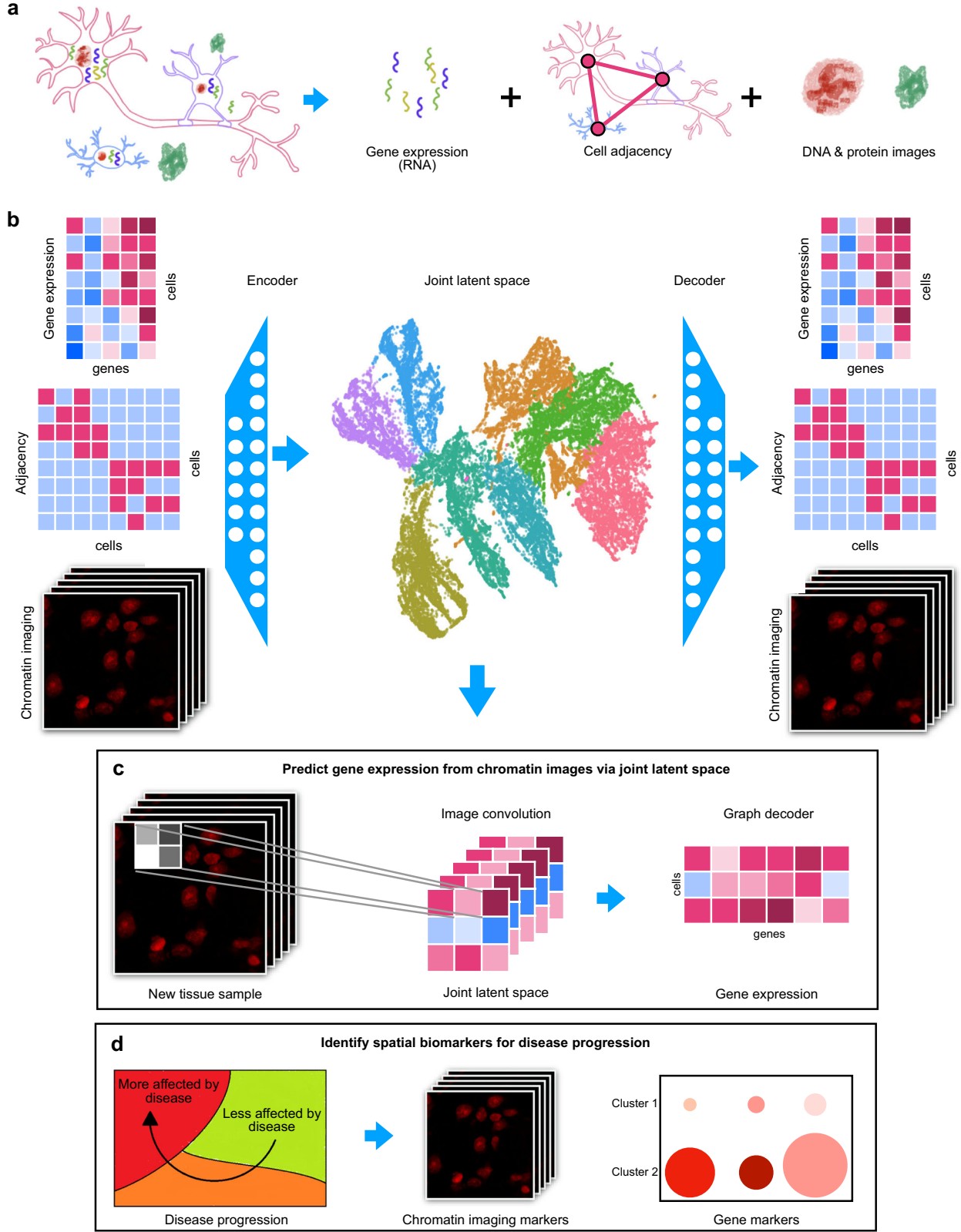

## Results

### STACI uses a graph-based autoencoder to obtain a joint representation of cell neighborhood and gene expression

Autoencoders are prominent neural network architectures that have been widely used for representation learning[17]. In the context of scRNA-seq, an autoencoder consists of two parts: the encoder learns a latent representation of each cell and the decoder reconstructs the cell's gene expression from its latent representation[19,20]. In its standard implementation, an autoencoder is trained to minimize the reconstruction error. Due to the sparsity in scRNA-seq data, instead of exact reconstruction, models have been explored that only estimate the parameters of a statistical model of gene expression[18–20]. To learn a

**Fig. 1 | STACI learns a joint representation of gene expression, cell adjacency, and chromatin images. a** STARmap PLUS jointly measures gene expression, chromatin condensation, and protein accumulation in whole tissue sections. Cell adjacencies are determined based on the location of the cell nuclei. **b** A joint latent space is computed for gene expression, cell adjacency, and chromatin images using an autoencoder neural network architecture. Separate decoders are used to reconstruct the three modalities from the joint latent space. UMAP is used to visualize the joint latent representation of all cells in the tissue samples; the cells are colored by cluster membership, with clustering performed in the joint latent space. **c** When a new tissue sample with only chromatin imaging is obtained, then the joint latent representation of this new sample can be inferred through the image encoder. Gene expression profiles of the new sample at single-cell resolution can be predicted from the joint latent representation using the trained gene expression decoder. **d** The joint latent space can be used to analyze the spatio-temporal progression of disease in a holistic manner using all available data modalities and to identify disease biomarkers including disease-associated changes in gene expression, cellular neighborhoods, nuclear morphology, and chromatin condensation.

joint latent representation of gene expression, cell location, and chromatin images, we build an autoencoder model that encodes the multiple modalities of a cell to a single latent representation. To ensure that the joint latent representation captures information from all modalities, we use a separate decoder to reconstruct each of the modalities from the same joint latent representation of each cell (Fig. 1b, Supplementary Fig. 1).

We first describe the subnetwork used to encode the spatial transcriptomics data (see below for integrating chromatin imaging data). To obtain a representation of gene expression that takes into account cellular neighborhoods, we use a graph representation of each tissue slice, where each cell is a node and an edge is placed between two nodes if the corresponding cells are in spatial proximity (Fig. 1a). The latent representation of a cell's gene expression profile informed by its location can then be obtained by using a graph-based convolutional encoder based on Kipf and Welling's definition, which performs a weighted average of a cell's gene expression vector with its neighbors' based on the edge weights in the graph[42–44]; see Supplementary Fig. 1. To ensure that both a cell's gene expression and its location are sufficiently captured in the latent representation, we use two separate decoders, one that ensures accurate reconstruction of gene expression and the other accurate reconstruction of cell adjacencies.

Our graph-based encoder approach leads to consistent and biologically meaningful tissue region annotations in the four STARmap PLUS samples of AD and control mouse brains (Fig. 2b). In particular, our model automatically segments the brain samples into continuous regions that correspond to the anatomical regions (cortex, hippocampus, dentate gyrus) described in the Allen Mouse Brain Atlas (Fig. 2f)[45]. Note that the resulting tissue segmentations are consistent across the four different mice, despite differences in tissue slicing, age, and disease states. Interestingly, our model identifies three clusters in the cortex (Fig. 2b, f), which correspond to the outer layers of the primary somatosensory area (cluster 1), inner layers of both the somatomotor area and the primary somatosensory area (cluster 2), and both the retrosplenial area and the outer layers of the somatomotor area (cluster 3). Although no data on amyloid plaque distribution was used in training our model, the three cortex clusters identified by our model show different distributions of amyloid plaques (Fig. 2g, h). Larger plaque sizes are observed in clusters 2 and 3 as compared to cluster 1 in both the 8-month AD mouse (p-values: 0.00042 for cluster 2 vs cluster 1 and 3.3e−20 for cluster 3 vs cluster 1) as well as the 13-month AD mouse (p-values: 0.0062 for cluster 2 vs cluster 1 and 1.7e−68 for cluster 3 vs cluster 1). We applied the trained STACI model and the same analysis to four new mouse samples held out for validation. All the resulting tissue segmentations are consistent in the new samples, including the separation into 3 clusters in the cortex regions with cluster 1 consistently having smaller plaques than the other two cortex clusters (Supplementary Fig. 2). This suggests that the cortex clusters obtained by our approach are disease-relevant and identify regions at different stages of AD progression.

In addition to over-parameterization discussed in the following section (Fig. 2b, c), the key ingredients of our graph-based autoencoder are its neural network architecture with separate decoders for gene expression and cell adjacency, the choice of the statistical model of gene expression, as well as the definition of cell adjacency used in the decoder. As discussed in the following paragraphs, each of these components is critical for obtaining consistent and biologically meaningful segmentation of tissue sections.

**Neural network architecture.** The graph autoencoder structure introduced by Kipf and Welling uses a single decoder that reconstructs node adjacencies[43], and was applied in the context of spatial transcriptomics in a previous study[46]. However, we find that using only a decoder for cell adjacency, in general, cannot sufficiently capture variations in gene expression. In fact, such an architecture cannot separate the hippocampal CA1 and CA2/CA3 regions, despite them being dominated by distinct cell types, and leads to inconsistent clusters in the cortex (Fig. 2e). Similar inconsistencies are observed if the adjacency decoder is removed instead (Supplementary Fig. 4c) or both decoders are removed leaving only the effect of graph convolution (Supplementary Fig. 4d). In contrast, our model with two separate decoders for gene expression and cell adjacency is able to recover the CA1 and CA2/CA3 regions (Fig. 2b). If cell locations are not used as input, the resulting clusters in the latent space mainly capture differences in gene expression stemming from different cell types and little spatial information, even when a cell adjacency decoder is used (Fig. 2d, Supplementary Fig. 4e). This variant of our model can be used for analyses focused on gene expression such as for cell type classification (Supplementary Fig. 5a).

**Statistical model of gene expression.** Instead of training the gene expression decoder of our model using the standard l2 reconstruction loss, we build on the Deep Count Autoencoder method[18] to model gene expression by predicting the parameters of a zero-inflated negative binomial (ZINB) distribution. In fact, given the sparsity of gene expression profiles in STARmap data, the standard l2 reconstruction loss is unable to reconstruct the observed gene expression profiles from the latent representations (Supplementary Fig. 3n). This suggests that a Gaussian distribution is unsuitable for modeling gene expression in STARmap datasets[17]. Modeling gene expression using other distributions, such as a negative binomial (NB) distribution, either results in inconsistencies with known anatomical structures of mouse brains or fails to recover consistent clusters in the cortex region (Supplementary Fig. 4f). In addition, using NB instead of a ZINB distribution results in a worse feature reconstruction loss (Supplementary Fig. 3i, m). Similar performance gains when using a ZINB distribution were observed also in the scRNA-seq setting[18].

**Definition of the cell adjacency matrix.** The definition of neighborhood of cells can be customized to the particular application. To demonstrate our model, we used a 20-nearest-neighbor adjacency matrix based on the Euclidean distance between the centroid of each cell in each sample to obtain the results in Fig. 2. Reducing the size of a cell's neighborhood from 20 nearest neighbors by half when defining the cell adjacency matrix reveals temporal dynamics and the impact of disease across samples: In the dentate gyrus (DG), cells separate into three clusters by samples, with DG of 8-months control and disease in

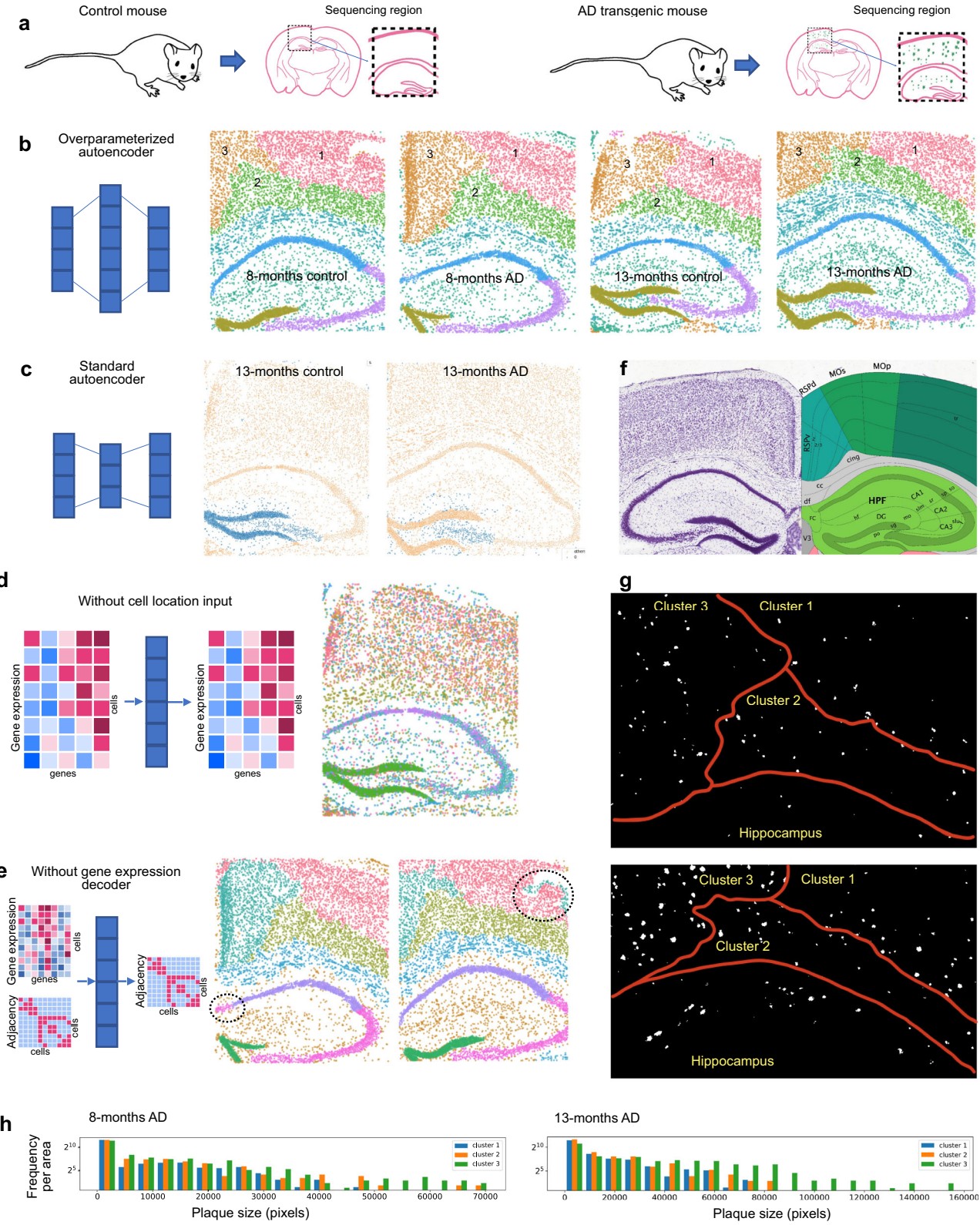

the same cluster (Supplementary Fig. 4a); interestingly, subclustering the spatial cluster corresponding to DG in our model using a larger neighborhood size recovers the same bifurcate trajectory from 8 months to 13 months (Supplementary Fig. 4b). In the cortex, consistent with our observation that the cluster 3 region indicates a more advanced disease stage (Fig. 2g, h), the model with smaller neighborhoods identifies differences in disease progression by separating the

cluster 3 region in the 8-months AD sample from the 13-months AD sample; in contrast, cluster 1, which indicates an earlier stage of AD, is dominated by age differences when using smaller neighborhoods (Supplementary Fig. 4a). Other definitions of cell adjacency/neighborhoods, for example based on a physical distance cutoff instead of k-nearest neighbors, can also be used depending on the particular application (Supplementary Fig. 3a, h, i, o).

**Fig. 2 | Over-parameterized graph-based autoencoder model enables consistent and biologically meaningful annotation of spatial tissue regions across multiple samples and tissue sections. a** The STARmap PLUS dataset[4] contains four mouse brain samples at 8 months and 13 months. Each time point consists of an AD mouse and a control mouse. Each tissue slice contains the cortex, corpus callosum, and hippocampus regions. **b** Clustering of the cells in the latent space learned by our over-parameterized autoencoder model leads to consistent spatial clusters across the four mouse samples. The latent dimension of this model is 6000 and a 20-nearest-neighbor graph was used to obtain the input cell adjacency matrix. Each dot is a cell plotted with its physical coordinates in the tissue and colored by the cluster memberships inferred in the latent space. **c** The use of standard (under-parameterized) autoencoder models leads to inconsistent spatial clusters across mouse samples: cells from the same cluster (blue) correspond to different regions in 13-months control (left) and 13-months AD (right) mice. The latent dimension of this model is 1024. **d** Clustering of the cells in the latent space

by our autoencoder model without cell adjacencies as input cannot separate different spatial neighborhoods. **e** Clustering of the cells in the latent space by our autoencoder model without the gene expression decoder fails to separate the CA1 region from the CA2/CA3 region and produces inconsistent spatial regions in the cortex across mouse samples. **f** A reference slice from the Allen Mouse Brain Atlas and Allen Reference Atlas - Mouse Brain[45] showing approximately the same anatomical region as in the STARmap samples. Our model automatically segments the brain samples into continuous regions that correspond to the reference anatomical regions (cortex, hippocampus, dentate gyrus). **g** Binary images of amyloid plaque in the cortex of the 8-months AD sample (top) and 13-months AD sample (bottom) showing the spatial differences in plaque distribution in the three cortex clusters identified by our model. **h** Histograms of plaque size, measured in number of pixels, plotted for the three cortex regions, indicate larger plaque sizes in clusters 2 and 3 as compared to cluster 1. Frequency is normalized by the area of each cortex region.

## STACI identifies consistent tissue regions across different samples through over-parameterization

Unlike standard autoencoders, which use lower-dimensional latent spaces for dimension reduction, STACI uses over-parameterized autoencoders, which represent the data in a higher dimensional latent space than the input space. The benefit of using over-parameterized autoencoders might be unintuitive since such networks have enough parameters to learn the identity map. However, over-parameterized autoencoders have been shown to self-regularize and lead to embeddings that stretch along the top principal components (PCs) of the data[28]. Thus, if the main sources of variation are biological, such as cell type differences, we hypothesize that over-parameterization can be used to emphasize the biological signal and reduce the sample-to-sample differences, thereby allowing for the integration of data from different tissue slices. In addition, over-parameterization is a simple approach that directly applies to any neural network model and avoids the use of separate methods to explicitly model batch effects[19,25,26].

Consistent with our hypothesis, encoding data from different tissue slices into a joint over-parameterized latent space yields clusters that are consistent across different samples (Fig. 2b). Namely, clusters in the latent representation correspond to the same tissue region across different samples with comparable fractions of cells in each tissue sample (Fig. 2b, Supplementary Fig. 6d). Note that the under-parameterized version of the same model results in strong sample-to-sample differences in the resulting clusters with different fractions of cells from each tissue sample (Supplementary Fig. 6a) as well cells in the same cluster corresponding to different spatial regions in different samples (Fig. 2c). Assessing batch effects using the average silhouette width, a measure used in previous studies[47–49], also indicates that over-parameterization leads to a significant reduction of batch effects (Supplementary Fig. 7a, Methods). This improvement in batch separation is further confirmed by using entropy of mixing, which improved from 0.79 in the under-parameterized model to 1.12 in the over-parameterized model and shows consistent improvements in each cluster (Supplementary Figs. 7a, Methods). The clusters were obtained by applying Leiden clustering[50] to the top 40 principal components of the latent representation with a clustering resolution of 0.1. Batch effect correction using an over-parameterized latent space is also observed when different Leiden resolutions are used (Supplementary Fig. 7b–d) and when different clustering methods are applied (Supplementary Fig. 7e, f, 8). Interestingly, the sample-to-sample variation in the under-parameterized model is less significant if cell location is not incorporated into the latent representation (Supplementary Fig. 6a, e), which indicates that the development of simple and effective methods for removing sample-to-sample variations is even more critical for the analysis of spatial transcriptomics data than what it already is for scRNA-seq data.

Consistent with our hypothesis, over-parameterization significantly increases the variance of the top PCs, and these do not correspond to sample-to-sample differences (Supplementary Fig. 6). Once the latent space dimension is increased sufficiently to remove sample-to-sample differences, the resulting embedding is insensitive to the exact number of latent dimensions being used (Supplementary Fig. 6b–d). An alternative neural network-based approach used standardly for batch effect correction in scRNA-seq data is to add an adversarial loss term in the latent space and train an additional neural network to penalize for any sample-to-sample differences[51]. While such an approach is able to reduce sample-to-sample differences also in the analyzed STARmap dataset, they are still present in some regions (Supplementary Fig. 9); over-parameterization achieves better performance without requiring additional neural networks or changing the training procedure.

Our approach is also applicable to other spatial transcriptomics technologies, such as the commercially available 10x Visium platform, and sequencing datasets beyond spatial transcriptomics, whenever the main sources of variation are biological and not given by batch effects or sample-to-sample differences. To demonstrate this, we applied STACI to a 10x Visium dataset of 12 mouse brain samples consisting of AD and control mice at different time points[52]. Compared to the clusters given by 10x based on gene expression alone (Supplementary Figs. 10a, 11), STACI achieves more consistent results across all samples, given the known anatomical regions of mouse brains (Supplementary Figs. 10c, 12). STACI also achieves better results in terms of consistency across samples and consistency with the known anatomical regions as compared to applying the same architecture with an under-parameterized latent space to input data batch corrected by mutual nearest neighbors (MNN)[48] or ComBat[53,54] (Supplementary Fig. 13). The computational resources required for the analysis of this dataset were recorded over six training epochs (Supplementary Fig. 14).

## STACI translates chromatin images to their corresponding gene expression profiles through the learned joint latent space

Current spatial transcriptomic technologies, such as STARmap and Visium, often obtain nuclear images together with spatial transcriptomic data in the same tissue section. Although these images contain rich information about cell type, epigenetic state, and the mechanical microenvironment of a cell, they are usually used only for pre-processing tasks such as cell segmentation or manual annotation of tissue regions, but not for downstream tasks[3,4,8]. In addition, current spatial transcriptomic methods are more expensive and time-consuming to obtain than DNA staining and it is thus of interest to develop methods that can translate from chromatin images to spatial transcriptomics.

To predict the gene expression profile of each cell, we use image patches centered at each cell of a diameter (15.14 μm) slightly larger

than an average cell size to ensure that the entire nucleus is contained. The image patches are embedded into the same latent space as the spatial transcriptomics data using a convolutional autoencoder that not only minimizes the reconstruction error of the image, but also the distance in the latent space between the transcriptomic and image representation of each cell (Supplementary Fig. 1, Method). The resulting joint latent space contains information from both spatial transcriptomics and chromatin images. When an unseen sample with only chromatin images is acquired, the sample can be embedded into the joint latent space by the image encoder and decoded to a gene expression profile, thereby enabling the translation from chromatin images to gene expression profiles. We here demonstrate the integration of images with only the chromatin channel to hold out the plaque images as an orthogonal validation of our analysis, but plaque and other multiplexed imaging channels (if available) can be incorporated into the joint latent space by using multi-channel cell images as input to the CNN autoencoder. Also unpaired datasets (such as scRNA-seq or scATAC-seq, if available) can be integrated without updating the existing autoencoders by training an additional autoencoder per new modality and choosing an appropriate distance metric, such as KL divergence[31], for matching the latent representation of the new modality with the existing joint latent representation.

To test our model, we omit the 8-month control sample from the training of both the graph-based autoencoder for the spatial transcriptomic data and the convolutional autoencoder for the chromatin images. We test our model by translating chromatin images of the 8-month control sample into gene expression profiles (Fig. 3a, Supplementary Figs. 15 and 16). Visualization of the resulting gene expression profiles by UMAP shows that the variation of the predicted gene expression profiles falls within that of the three training samples (Fig. 3c). Clustering the predicted gene expression profiles of all samples and comparing the expression of cluster markers to a reference atlas[53] shows that the clusters correspond to cell types (Fig. 3c, d, Methods), that the proportion of different cell types are consistent across samples (Supplementary Fig. 17c), and that the cluster identities as well as the identified cell type markers are consistent with a previous study[4] (Fig. 3e, Supplementary Fig. 17d). Although our model is not optimized for predicting gene expression of single cells, it is able to translate chromatin images to gene expression profiles in unseen samples and generalize to new experimental conditions, thereby indicating that the joint latent space captures functional information in both gene expression and chromatin organization.

## Chromatin condensation is predictive of the size of amyloid plaques in a cell's neighborhood

With the joint latent space providing a joint representation of both spatial transcriptomics and chromatin images, we can study how disease progresses in different regions of the tissue and connect the disease mechanism to both nuclear morphology and gene expression. To do this, we train a fully connected neural network on the joint latent space to predict the size of plaques in an image patch, which encompasses a similarly sized neighborhood as the 20-nearest-neighbors used to form the adjacency matrix for the spatial transcriptomics autoencoder, centered at each nucleus (Fig. 4a, b, Supplementary Fig. 18b). We define a cell as positive if there is plaque in the image patch centered at the cell. Applying our regression model to this classification task using a cutoff in the predicted plaque size is able to generalize to all cortex regions in the 8-month and 13-month control mice (Supplementary Data 1, model #18). In addition, it can differentiate the positive and negative cells of cluster 1 in the AD mouse at both 8 and 13 months (Supplementary Data 1, model #18). This suggests that the joint latent space captures the changes in cells in response to nearby plaques and that our model

is able to identify such changes. But despite the good generalization accuracy across various clusters and samples, our model is unable to differentiate the positive and negative cells in cluster 3 in the 13-month AD mouse, even when using them for training (Supplementary Fig. 18c, Supplementary Data 1). This indicates that cells further away from plaques in cluster 3 have similar gene expression, cell neighborhood, and nuclear morphology as cells close to plaques (within the input image patch). Indeed, when plotting the classification predictions by cell location in the tissue samples, we observe that while cells classified as positive are mostly within the input image patches containing plaques in cluster 1, positive classifications also appear at larger distances from plaques in cluster 3 (Supplementary Fig. 18d). This observation further supports our hypothesis that cluster 3 corresponds to a cortex region that is more advanced in the disease progression as compared to cluster 1. A similar analysis also suggests that cortex cluster 2 is more advanced than cluster 1 with respect to disease progression (Supplementary Fig. 18d, Supplementary Data 1).

Next, we examine which chromatin and gene expression features are used by our regression model for predicting plaque size. To determine the features in the joint latent space that contribute positively to the plaque size prediction, we use gradient backpropagation from the regression output to the latent features. Similar to the Grad-CAM method[54], the latent feature activations are then used to map the last convolutional layer of the image encoder back to the input chromatin images (Methods; Fig. 4c). This results in a value (the gradient) per pixel in the chromatin images, which indicates the predictiveness of the pixel for plaque size. By segmenting the nuclear images (Methods, Supplementary Fig. 20) and computing the average gradient in each nucleus, we find notable cell-type specific differences; microglia in the AD mice, for example, have a higher gradient at both time points than excitatory or inhibitory neurons, indicating that they are more predictive of plaque size (Fig. 4d, Supplementary Fig. 21).

A more careful inspection of the regression gradient in each cell indicates that the chromatin features used for plaque prediction are of subcellular scale and may be associated with chromatin condensation (Fig. 4e–g). Changes in chromatin condensation patterns have previously been found to be associated with mechanical signals from the microenvironment[8,18]. Interestingly, the distribution of chromatin intensity increases with disease progression in all cell types, and the distribution in non-neuronal cells tends to become bimodal, thereby indicating more pronounced euchromatin and heterochromatin states (Figs. 4e, 5a, Supplementary Figs. 24, 25). We summarize the chromatin condensation state of a cell by its heterochromatin ratio (total chromatin pixel intensity of heterochromatin regions in the cell normalized by total chromatin pixel intensity of the cell), where heterochromatin regions are defined via a threshold on chromatin intensity established in a prior study[18]. While the threshold was established in fibroblasts, it is consistent with the natural cutoff in the bimodal distributions that we observe (Figs. 4e, 5a, Supplementary Fig. 24). Similar to the gradient of plaque size regression, heterochromatin ratio exhibits differences between cell types, e.g. microglia and oligodendrocytes have higher heterochromatin ratios than excitatory and inhibitory neurons, and the heterochromatin ratio increases significantly with disease across nearly all cell types (Fig. 4f, Supplementary Fig. 22). This is particularly notable given the general trend that chromatin decondenses with aging[55], which manifests itself through a decrease in chromatin intensity and heterochromatin ratio from 8 to 13 months (Supplementary Figs. 22, 26, 27). Interestingly, the heterochromatin ratio shows a strong association with the gradient of plaque size regression in all cell types (Fig. 4g, Supplementary Fig. 23). Thus, chromatin condensation, as measured by the heterochromatin ratio, is indicative of disease progression at single-cell resolution within the tissue microenvironment.

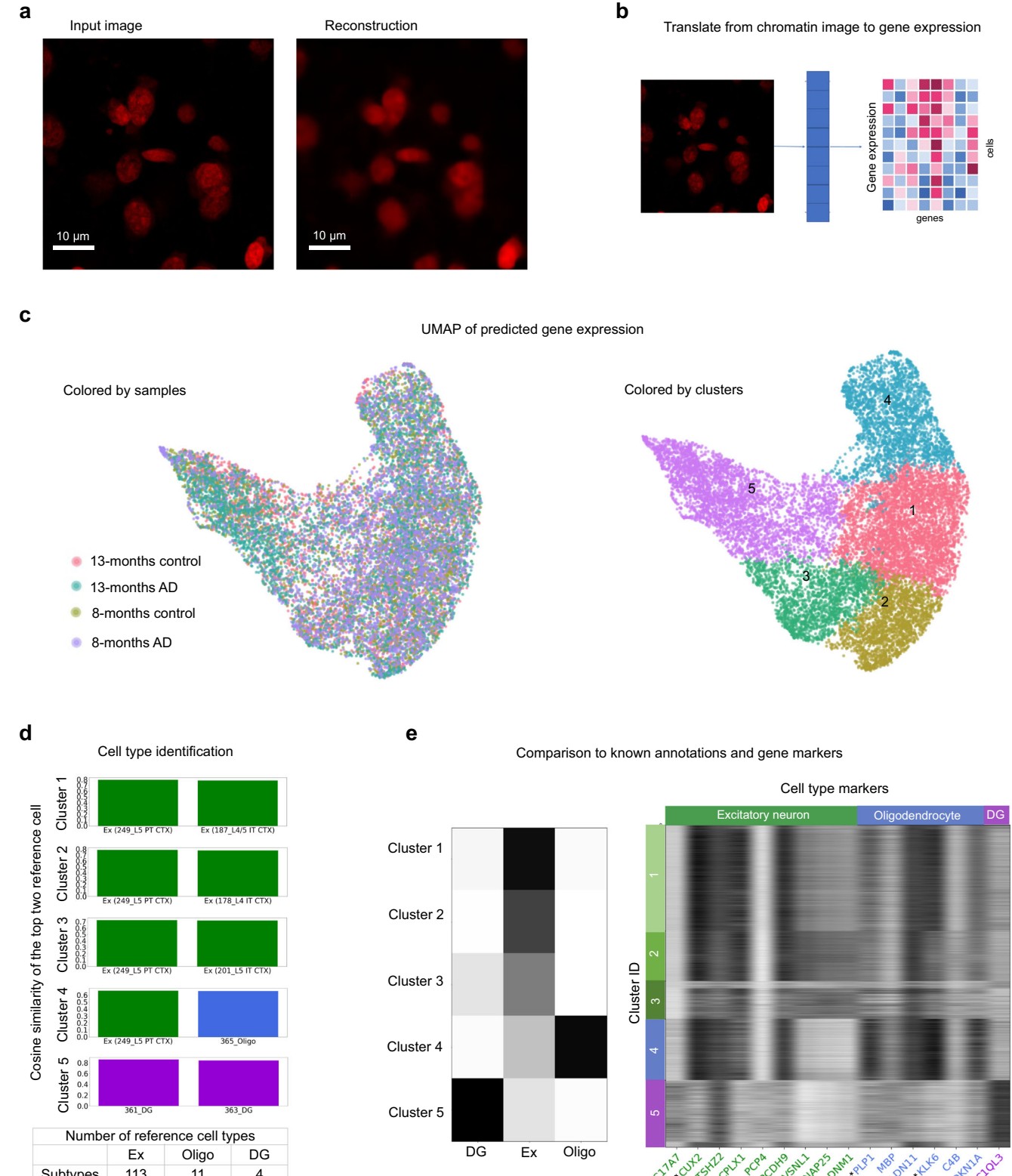

## Nuclear morphology and gene expression are associated with cell type-specific spatio-temporal disease response

Characterizing cells by chromatin condensation and regression gradient has the potential to allow the identification of spatio-temporal changes in AD. Indeed, the spatial distribution of chromatin condensation is in line with our previous observation that cortex clusters 2 and 3 are more advanced in disease than cluster 1—the chromatin intensity and heterochromatin ratio in cluster 1 are lower than in

clusters 2 and 3 for all cell types in both the 8-months and 13-months AD samples (Fig. 5a, Supplementary Figs. 22, 24, 25). To analyze whether changes in regression gradient and heterochromatin ratio are also associated with changes in subtype composition, we divide cells of each cell type into four quadrants by setting thresholds for regression gradient and heterochromatin ratio (Fig. 5a, Supplementary Figs. 24, 25). For all cell types, the fraction of cells with high heterochromatin ratios and high gradients (top right quadrant), is larger in cortex

**Fig. 3 | STACI translates chromatin images to gene expression profiles through the learned joint latent space.** The 8-months control sample was not used in training the graph or image autoencoders. **a** An example of the reconstruction of a chromatin image in the 8-months control sample using the image autoencoder. Similar reconstruction quality of chromatin images is observed across the 8-months control sample with 8062 cells. **b** Schematic of translating chromatin images to gene expression. The joint latent space embedding is inferred from the chromatin images by the image encoder. Single-cell gene expression profiles are predicted from the inferred joint latent space by the gene expression decoder. **c** UMAP of the predicted gene expression profiles. The predicted single-cell gene expression profiles of the test sample (8-months control) falls within the variation of the training samples (left). Leiden clustering applied to the predicted gene expression profiles identifies five clusters (right). **d** Bar plot of average cosine similarity between cells in each of the five clusters and the mean expression of the top two cell types in a reference atlas[53] (ranked by average cosine similarity). The lower table shows the number of reference cell types in the atlas corresponding to excitatory neurons (Ex), oligodendrocytes (Oligo), and dentate gyrus (DG). **e** Cell type composition in each cluster in the 8-months control sample as annotated in Zeng et al.[4] (left) is consistent with the predicted cell type annotations in (**d**). Predicted expression of known cell type markers in each cluster in the 8-months control sample excluded from training (right) show cell type specificity. *Differentially expressed cluster markers for each cell type; $p$-val < 0.05 and fold change > 1.1. Excitatory neurons (excluding CA1, CA2, CA3), DG, and oligodendrocytes in the corpus callosum are included in the clustering to represent major neuronal and glial cell types. All samples were used for clustering and differential expression analysis.

clusters 2 and 3 as compared to cluster 1, in line with our hypothesis that the cortex regions corresponding to clusters 2 and 3 are at a more advanced disease state. Looking more closely at microglia, we observe an increase in their number in each cluster of the two AD samples compared to the controls (Fig. 4e, Supplementary Fig. 28); in fact, the proliferation of microglia is known to accompany microglia activation in AD[56,57]. Looking more closely at the different subtypes of microglia (M1-M3)[4] reveals that in cortex cluster 1 of the 8-month AD sample, which we hypothesize to be at the earliest stage of AD, the known disease-associated subtype of microglia (DAM, M3) has high heterochromatin ratio and high regression gradient, whereas the other subtypes dominate the lower left quadrant with low heterochromatin ratio and low gradient (Fig. 5a). As the disease progresses in cluster 2, we observe that the non-disease subtypes (M1 and M2) start to show high chromatin condensation and regression gradient. Finally, the cluster 3 region of the cortex shows a large number of DAM cells, a majority of which have high gradient and high heterochromatin. This suggests that healthy subtypes of microglia undergo chromatin condensation and become predictive of nearby plaques before differentiating into the DAM subtype and changing their gene expression profiles.

Given our observation that the regression gradient and heterochromatin ratio are associated with functional properties of disease progression including subtype composition, we hypothesize that these features can be used to obtain more powerful and more specific gene markers of disease. We identify such disease markers through a differential expression analysis (DE) of AD cells in the high-gradient high-heterochromatin quadrant and control cells in the low-gradient low-heterochromatin quadrant. In microglia, such an analysis identifies the two known marker genes of the DAM subtype, CD9 and CST7, to be strongly associated with the cluster 3 region in the cortex, which are not identified by a standard differential expression analysis of microglia that are found in spatial proximity to plaques versus further away (Fig. 5b, Supplementary Fig. 19). Similarly, in oligodendrocytes, we identify a disease gene marker PPP1R9B and three myelin-associated genes, MOBP, JPH4, and PLP1 to be strongly associated with clusters 2 and 3, which are not identifiable using a standard differential expression analysis (Fig. 5b, Supplementary Fig. 19). Similarly, in excitatory neurons, there is a significant increase in the number of discovered marker genes in the 8-months AD sample when thresholding by heterochromatin ratio and gradient, compared to a standard differential expression analysis of diseased versus control (Fig. 5b, Supplementary Fig. 19). These marker genes also show a consistent spatial distribution across the three cortex regions we identified, e.g. CTSB is increasingly upregulated in microglia from cluster 1 to cluster 2 and 3 (Supplementary Fig. 29). These marker genes are enriched in gene ontology terms related to synaptic transmission, secretion, protein localization, and transport (Fig. 5c), which have known associations with AD[36,58,59]. Consistent with the association between chromatin condensation and disease as an indication of the change in the mechanical microenvironment, six of the marker genes are annotated for regulation of cell size and/or cellular component size (KCNMA1, PAK1, DBN1, PFN2,

SEMA3C, CFL1), in addition to marker genes involved in transmembrane signaling and cellular component organization. There is a loss of excitatory neurons in cortex cluster 3 of the 13-months AD sample, compared to both the 13-months control and the 8-months AD samples (Supplementary Fig. 28), which is another indication that cluster 3 is ahead in disease progression as compared to the other cortex clusters. Differential expression analysis identifies genes that can mostly be attributed to the subtype Ex2, which is the dominant subtype in cluster 3 and shows strong signs of cell death (Fig.5b, Supplementary Figs. 25, 27). While FAIM2, an anti-apoptotic gene, is also identified by our analysis, most genes including DDIT3 are associated with cell death, signaling, and synaptic transport[60,61]. Our analysis demonstrates that chromatin condensation together with our regression model is capable of identifying spatio-temporal changes in disease as well as disease gene markers.

## Discussion

We presented STACI, a framework for integrating multi-modal spatial data with built-in batch correction. Current strategies that analyze spatial transcriptomics data using scRNA-seq methods ignore the spatial context of cells, although biological processes often involve changes in the spatial organization of cells[1,2,62]. Analogous to image convolution, where taking the neighborhood of each pixel into account is critical for the performance of downstream tasks, we introduced a graph convolutional autoencoder that integrates both the gene expression of a cell and that of its neighbors. Our graph-based autoencoder structure decodes both a cell's gene expression profile as well as its adjacencies. Unlike when using other graph convolutional methods[43,46], clustering cells in the latent space embedding obtained by STACI leads to segmentations of the tissue sections into known anatomical regions. In addition, we proposed the use of overparameterization as a simple and effective strategy to integrate different samples into the same latent space and showed that this results in consistent clusters across different tissue samples, despite the gene expression and morphological differences in the tissue slices. Batch correction methods developed for scRNA-seq do not take tissue morphology into account and are ineffective when both gene expression and cell location are used in the analysis. STACI is applicable in this situation and provides a simple batch correction approach through over-parameterization that retains the neural network architecture and does not require specifying a statistical model of batch effects. Besides the separation into known anatomical regions, such as cortex and corpus callosum, our model separates the cortex into three regions. Interestingly, our analysis suggests that this separation is disease-relevant, with the three regions showing differences in number and size of amyloid plaques as well as gene expression and chromatin condensation states of cells.

STACI provides a framework for integrating additional data modalities with spatial transcriptomics data. In particular, we explored the integration of spatial transcriptomics with chromatin imaging. A number of studies have revealed that chromatin organization reflects

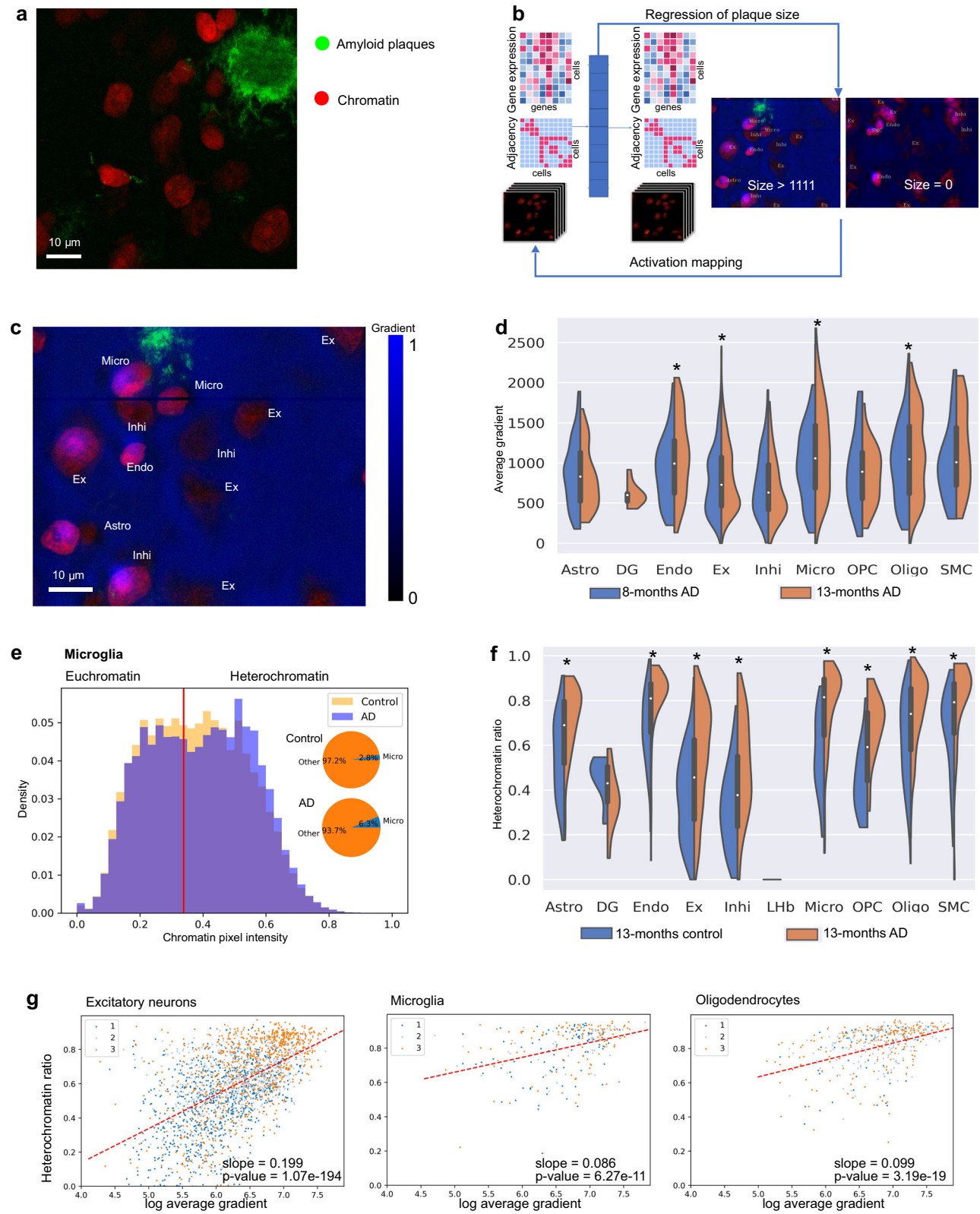

the mechanical microenvironment of a cell and is critical for the co-regulation of genes[29,63–67]. Various cell types are mechano-sensitive and mechano-chemical signals are known to be associated with the progression of different diseases[63–67]. It is conceivable that the accumulation of amyloid plaques in AD changes the mechanical microenvironment, but little is known about its effect on neighboring cells. In order to link the physical and biochemical space, STACI simultaneously represents gene expression, cell location, and chromatin features in a joint latent space. By associating this joint latent space with amyloid plaques, we found that chromatin condensation of cells is predictive of the size of nearby plaques (as measured by regression gradients). Interestingly, STACI also allows incorporating

**Fig. 4 | Prediction of amyloid plaque size from the joint latent representation results in the identification of cell types and chromatin features that are predictive of plaque near cells. a** An example of overlaid chromatin (red) and amyloid plaque (green) channels from the 13-months AD sample with 10021 cells. **b** Schematic of our regression model that predicts plaque size from the joint latent representation of gene expression, cell adjacencies, and chromatin images. **c** Regression gradient (blue) projected to the input chromatin image (red). The analysis uses the two AD mouse samples with at least 7257 cells each. **d** Distribution of regression gradients within single cells in each cell type in cluster 3 of AD samples. *$p$-value < 0.05 for 8-month vs 13-month samples. Cells are from two mice with at least 7257 cells each. Bounds of boxes indicate quartiles. Whiskers indicate extrema. Two-sided $T$-tests are performed and raw $p$-values are 0.2446 (Astro), 0.002677 (Endo), 0.04300 (Ex), 0.2872 (Inhi), 0.002549 (Micro), 0.6968 (OPC), 0.01780 (Oligo), 0.1019 (SMC). **e** Histogram of chromatin pixel intensities of

microglia in the cortex of the 8-months samples (normalized to sum to 1, since the proportion of microglia increases in AD; see inset). Red line: threshold of heterochromatin pixels (see Methods). **f** Distribution of heterochromatin ratio of each cell type in cluster 3 of 13-months control and AD samples. *$p$-value < 0.05 for control vs AD samples. Cells are from two mice with at least 7766 cells each. Bounds of boxes indicate quartiles. Whiskers indicate extrema. Two-sided $T$-tests are performed and raw p-values are 8.715e−9 (Astro), 0.07304 (DG), 2.705e−8 (Endo), 1.380e−41 (Ex), 9.950e−8 (Inhi), 6.053e−8 (Micro), 1.111e−4 (OPC), 4.544e−13 (Oligo), 9.479e−4 (SMC). **g** Heterochromatin ratio versus $\log_2$ average gradient of cells in the cortex of the 8-months AD sample colored by cortex cluster membership. Red line fitted using linear regression. $P$-values are calculated by two-sided Wald Test and the null hypothesis that the slope is 0. Ex excitatory neuron, Micro microglia, Inhi inhibitory neuron, Endo endothelial cell, Astro astrocytes, OPC oligodendrocyte precursor cell, SMC smooth muscle cell.

new samples where only a single modality is available to predict the missing modality and e.g. translate between chromatin imaging and spatial transcriptomics, as well as perform downstream tasks in the joint latent space given just one modality.

While we demonstrated the use of STACI on STARmap PLUS data, it can be directly applied to data from other spatial transcriptomics or proteomics technologies including 10x Visium, Slide-seq, MERFISH, seqFISH, and CODEX[5,6,8,68]. Cells/beads/spots in these datasets are represented as nodes in a graph, and adjacencies in the graph can be customized to represent a fixed-size neighborhood with a distance threshold or to capture density changes with k-nearest neighbors. To ensure over-parameterization, the latent dimension of the model should be larger than the input feature dimension, i.e. the number of genes or proteins, but the model is not sensitive to the exact choice of the latent dimension. We demonstrated the integration of different data modalities based on paired spatial transcriptomics and imaging data obtained in the same tissue section by using an l2 loss to match the different modalities in the latent space. It is possible to extend our method to datasets where the spatial transcriptomics and imaging data are not obtained in the same tissue section, such as in Slide-seq, by using a discriminative loss[31]. Collectively, we presented a method for analyzing disease progression in complex tissue microenvironments by combining multiple data modalities, thereby allowing the identification of disease biomarkers that capture gene expression combined with cell location and chromatin condensation patterns.

## Methods
### Ethical statement
All animal procedures followed animal care guidelines approved by the Genentech Institutional Animal Care and Use Committee (IACUC) and animal experiments were conducted in compliance with IACUC policies and NIH guidelines[4].

### Graph convolutional autoencoder
We introduced a graph convolutional autoencoder to compute a joint representation of gene expression and cell adjacency. A graph G = (V, E) is constructed for each tissue sample. Each node $v_i$ is a cell and its feature is the gene expression of cell i. An edge is added between two cells given a user-defined distance threshold for spatial proximity. We tested k-nearest neighbors by Euclidean distance and physical distance thresholds (Supplementary Fig. 3). The input feature matrix X contains the gene expression of all cells in the sample and is of size N × D, where N is the number of cells and D is the number of genes. The input adjacency matrix $A_{N \times N}$ is binary in our paper, but can be weighted by a function of the distance between cells based on the particular application of interest. Our model builds on variational graph autoencoders[43], using the same graph encoder and adjacency matrix decoder. The graph encoder consists of two graph convolutional layers and computes a latent feature $z_i$ of size F for each cell i. The

adjacency matrix decoder calculates a reconstructed adjacency matrix $\widetilde{A}$, such that $\widetilde{A}_{ij} = sigmoid(z_i^T z_j)$ is the reconstructed edge weight between cells i and j. We added a gene expression decoder that decodes the dropout rate $\pi$, mean $\mu$, and dispersion $\theta$ of a zero-inflated negative binomial (ZINB) distribution[18] from the latent space of the graph encoder that maximizes the likelihood of the input gene expression. The inferred parameters of the ZINB distribution have the same size N × D as the input gene expression matrix and are defined as:

$$H = Leaky ReLU(ZW_0) \tag{1}$$

$$\pi = sigmoid(HW_\pi) \tag{2}$$

$$\mu = \exp(HW_\mu) \tag{3}$$

$$\theta = \exp(HW_\theta) \tag{4}$$

Alternatively, we tested a variant of our model, in which the gene expression decoder predicts the parameters of a negative binomial (NB) distribution, i.e. omitting the dropout rate $\pi$. In training, we simultaneously minimize the reconstruction loss of the two decoders and the Kullback-Leibler (KL) divergence between the latent distribution Z and a Gaussian prior:

$$L = Binary CrossEntropy(\widetilde{A}, A) - \alpha^* \log[ZINB(X; \pi, \mu, \theta)] + \beta^* KL[q(X, A)|p(Z)], \tag{5}$$

where $\alpha$ and $\beta$ are hyperparameters, and $q(X,A)$ and $p(Z)$ correspond to the inference model and Gaussian prior defined previously[43]. $ZINB(X; \pi, \mu, \theta)$ is replaced by $NB(X; \mu, \theta)$ in the NB model. Leaky ReLU activation[69] is used for all hidden layers. In training, we omitted 15% of randomly selected nodes and their corresponding edges; 5% of the omitted nodes were used for validation and the remaining 10% were used for testing. All models were trained using the Adam optimizer[17] with a learning rate of 0.001.

**Clustering and visualization of the latent features.** All clustering was performed using the Leiden clustering method implemented in the SCANPY package[24,50]. We followed the standard procedure for Leiden clustering of single-cell data by computing a neighborhood graph from the top 40 principal components of the latent features and clustering the neighborhood graph[24]. For the neighborhood graph, the neighborhood size, "n_neighbors", was set to 10. The resolution of Leiden clustering was set to 0.1 to obtain the clustering in Figs. 1b and 2b. The visualization of latent features of cells using UMAP[70] also

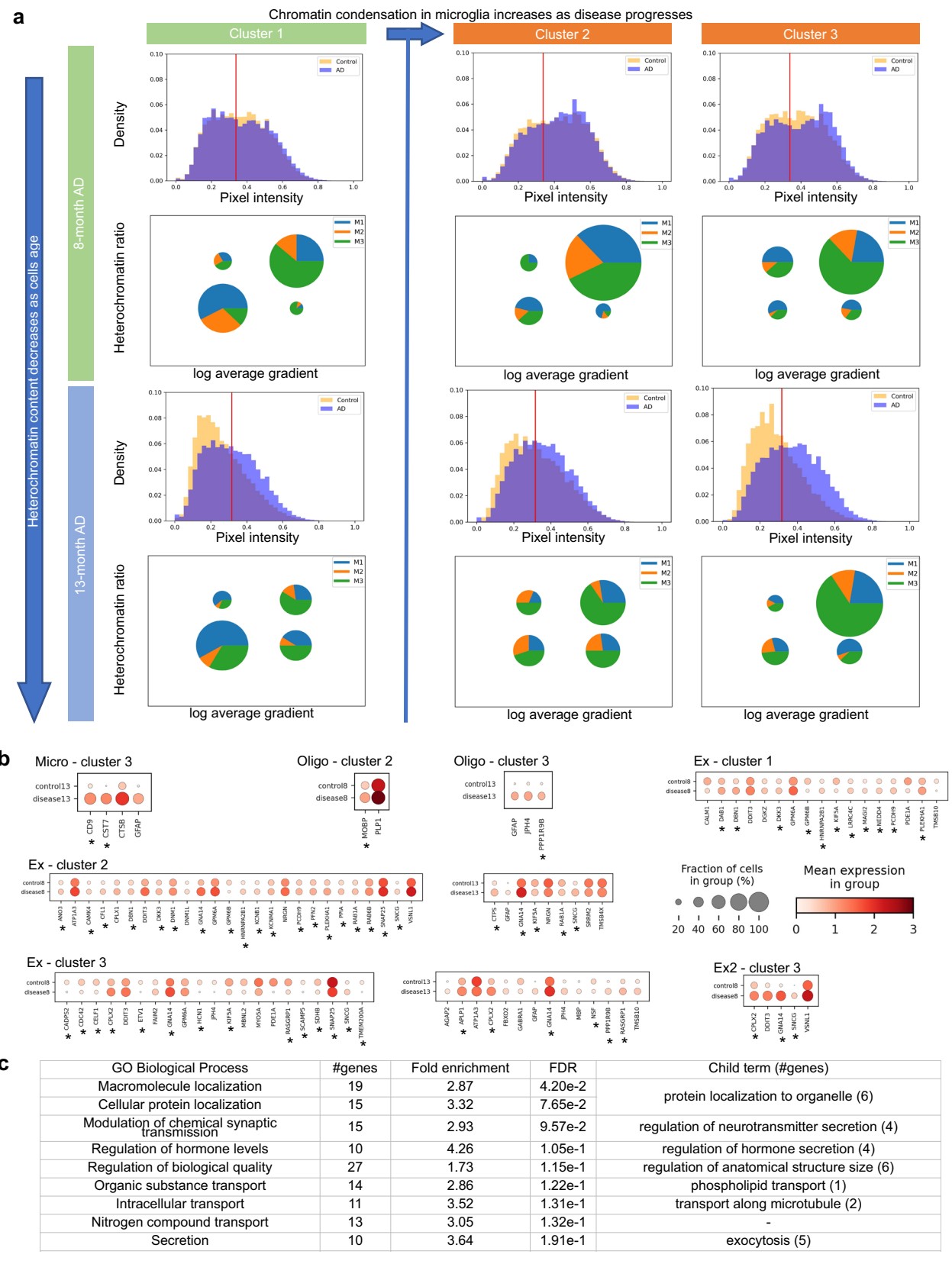

**a**

**c**

| GO Biological Process | #genes | Fold enrichment | FDR | Child term (#genes) |
|---|---|---|---|---|
| Macromolecule localization | 19 | 2.87 | 4.20e-2 | protein localization to organelle (6) |
| Cellular protein localization | 15 | 3.32 | 7.65e-2 | |
| Modulation of chemical synaptic transmission | 15 | 2.93 | 9.57e-2 | regulation of neurotransmitter secretion (4) |
| Regulation of hormone levels | 10 | 4.26 | 1.05e-1 | regulation of hormone secretion (4) |
| Regulation of biological quality | 27 | 1.73 | 1.15e-1 | regulation of anatomical structure size (6) |
| Organic substance transport | 14 | 2.86 | 1.22e-1 | phospholipid transport (1) |
| Intracellular transport | 11 | 3.52 | 1.31e-1 | transport along microtubule (2) |
| Nitrogen compound transport | 13 | 3.05 | 1.32e-1 | - |
| Secretion | 10 | 3.64 | 1.91e-1 | exocytosis (5) |

used a neighborhood size, "n_neighbors", of 10 and "min_dist" was set to 0.25.

**Metrics of batch effect correction.** The silhouette score is computed using the silhouette_batch method in the scib package[49]. The entropy of mixing follows the implementation in a previous study[48]. Given

100 randomly selected cells from all four samples, the entropy of each cell is computed given the sample label of the 50 nearest cells in the latent space. The average entropy over the 100 randomly selected cells was averaged over 100 bootstrap samples. The entropy of mixing is calculated for all cells and also for cells in each cluster separately.

**Fig. 5 | Chromatin condensation is indicative of the impact of amyloid plaques on cells. a** Chromatin intensity, heterochromatin ratio, and regression gradient of plaque size in microglia. The known disease-associated subtype of microglia (DAM, M3) has high heterochromatin ratio and high regression gradient, whereas the other subtypes dominate the lower left quadrant with low heterochromatin ratio and low gradient. Histograms: chromatin pixel intensities of all microglia in the three cortex clusters at each time point, normalized to sum to 1 (see Methods). Red line: threshold for identifying heterochromatin pixels (see Methods). Pie charts: microglia grouped into four quadrants by heterochromatin ratio (threshold at 0.8) and log average gradient (threshold at 6.7). Size of a pie chart is proportional to the fraction of cells in the respective quadrant. Angles in a pie chart are proportional to the fraction of each cell subtype in the respective quadrant. **b** Using regression gradient and heterochromatin ratio to obtain more powerful and specific transcriptomic markers of disease. Differential expression analysis of cells with high gradient and high heterochromatin ratio (top right quadrant) in the AD samples compared to cells with low gradient and low heterochromatin ratio (lower left quadrant) in the control samples at the same time points. The threshold of differential expression using Wilcoxon rank-sum test[79] is *p*-value <0.05 after adjustment by Benjamini-Hochberg procedure[73] and fold change of at least 10% in either direction. At most 25 upregulated and 25 downregulated genes with the smallest *p*-values are shown. *Genes not found in the standard DE analysis between cells close to and far away from plaques (see Supplementary Fig. 19). The exact *p*-values and log fold changes of all differentially expressed genes are provided in Source Data 1. **c** Gene ontology enrichment analysis of upregulated differentially expressed genes in excitatory neurons in cortex cluster 2 of the 8-months AD sample are related to synaptic transmission, secretion, protein localization, and transport, which have known associations with AD. The background of the enrichment analysis consists of all genes in the STARmap PLUS dataset.

## Joint latent space of spatial transcriptomics and chromatin images

A standard variational CNN autoencoder[17] was trained on 2D chromatin images obtained through maximum projection of the 3D chromatin images. Each input image patch is a d × d square centered at the centroid of a cell and min-max scaled to [0, 1]. When the joint latent space is used for predicting gene expression of single cells in a new sample with only chromatin images, we set d to 15.14 μm, which is slightly larger than the diameter of a cell allowing the prediction of the gene expression to focus on the target cell. When the joint latent space is used for downstream analysis, such as for the regression of plaque sizes, d was set to 75.68 μm, which is comparable to the neighborhood size of the 20-nearest-neighbors cell adjacency used in the graph convolution. The CNN encoder has five convolutional layers, followed by two fully connected layers to separately compute the mean and dispersion from which the latent features are sampled. The decoder is the inverse of the encoder. This is similar to the CNN autoencoder in our earlier work[31]. Leaky ReLU activation[69] was used for all hidden layers. We trained and fixed the parameters of the graph autoencoder before training the CNN autoencoder. The latent space Z′ of the CNN has the same dimension as the latent space Z of the graph autoencoder. The training loss of the CNN autoencoder is the sum of the CNN reconstruction loss calculated by the standard $l_2$ loss and the $l_2$ loss between Z and Z′. The same split of training and validation as for the graph autoencoder training was used. To further test the model's ability to predict gene expression of unseen samples, we omitted the entire 8-month control sample from training both the graph autoencoder and the CNN autoencoder. To test the incorporation of unseen samples in the joint latent space for the downstream regression, both 8-month control and AD samples were omitted in training the graph and CNN autoencoders. When incorporating an unseen sample, the CNN encoder was used to infer the joint latent space from the chromatin images. The gene expression decoder of the graph autoencoder was used to predict the gene expression from the inferred latent space. The mean μ of the ZINB distribution was used as the predicted gene expression. The average negative log ZINB likelihood loss of the gene expression predicted from DNA images is comparable to the reconstructed gene expression when gene expression and cell locations are used as the input to the graph convolutional autoencoder (prediction loss = 1.3969; reconstruction loss = 1.3725).

## Cell type annotation using predicted gene expression

We used a reference gene expression dataset[53] to annotate a mixture of three representative neuronal and glial cell types by their predicted gene expression: oligodendrocytes in the white matter, DG, and cortical excitatory neurons. Five clusters were obtained from the predicted gene expression by Leiden clustering[50] (resolution = 0.2). Cluster markers of each cluster compared to all other cells outside of the cluster were obtained by differential expression analysis using the predicted gene expression. The union of upregulated genes in each cluster was used for the following comparisons. For each subtype of oligodendrocytes, DG, and cortical excitatory neurons, we calculated the cosine similarity between its mean gene expression in the reference dataset and the predicted gene expression of each cell in our dataset. The cosine similarity of each reference subtype was averaged across all cells in the same cluster.

## Adversarial loss in the latent space

In addition to the over-parameterization approach for batch effect removal, we explored the use of an adversarial loss in the latent space by training an additional discriminator. The discriminator was trained to assign the correct sample labels to cells from each sample given the latent representation of cells as the input. For the discriminator we used a fully connected network with one hidden layer of size 128 and leaky ReLU activation[69]. For the loss of the discriminator we used the cross-entropy loss between the true sample label and the discriminator output after sigmoid activation. In order to train the graph autoencoder to make the discriminator output equal probabilities of all sample labels for any given input sample, an additional adversarial loss term was added to the loss of the graph autoencoder. For this adversarial loss we used the cross-entropy between the discriminator output after sigmoid activation and the vector $[0.5, 0.5, 0.5, 0.5]^T$, using the discriminator updated in the previous epoch. The target vector represents a cell having equal probability of being in any of the four tissue samples. We alternated the training between the under-parameterized graph autoencoder (1024 dimensions) and the discriminator, such that only one model was updated in each epoch and the other model's parameters remained fixed.

## Spatial transcriptomics data pre-processing

We obtained the raw count matrix of gene expression and the spatial location of cell centroids from Zeng et al.[4] The 2112 genes, which passed filtering in Zeng et al.[4], were used in our analysis. For the graph autoencoder, the raw gene expression $\bar{x}_i$ of cell i was normalized as $x_i = min\_max(log_2(\bar{x}_i + 0.5))$ to a range of [0, 1]. This normalization method achieved lower gene expression reconstruction loss in validation and testing than scaling the expression of each gene across all cells by z-score normalization (Supplementary Fig. 3i, p). For differential expression analysis, the input gene expression in each cell was normalized by the total number of gene counts in the cells, following Zeng et al.[4].

## Experimental setup for imaging

All images were obtained with a Leica TCS SP8 confocal microscope and with a 40× objective. The voxel size is 0.0946 × 0.0946 × 0.3463 μm³ (x, y, and z dimensions respectively). Propidium Iodide (PI) staining was applied according to the manufacturer's protocol. A detailed protocol of the experiment can be found in the STARmap PLUS paper[4].

### 3D segmentation of chromatin images

We used 3D chromatin images to obtain nuclear features and associated these features to plaque size using the regression gradient. The python package py-clesperanto of CLIJ was used for the 3D segmentation of chromatin images[71]. For each cell, we used all the z-stack images and cropped the horizontal directions to $37.84 \times 37.84\ \mu m^2$ centered at the cell centroid. After min-max scaling to [0, 1], the 3D stack of each cell was further cropped to $18.92 \times 18.92\ \mu m^2$ in the horizontal directions. The images were resampled in the z-direction to have isotropic voxels. Then we applied Gaussian blur, spot detection, a second Gaussian blur with sigma set to 3, Otsu thresholding, and Voronoi labeling. The first Gaussian blur was optimized with two iterative searches to obtain the maximum sigma value for which a cell can be detected at the given centroid. We used "binary_fill_holes" in the SciPy package with a $2 \times 2 \times 3$ matrix of ones as the structuring element on the resulting mask after Voronoi labeling[72].

### Regression of plaque size

We trained a fully connected network with three hidden layers of size 1024 to predict the size of plaque near a cell given the cell's latent representation. The input was either the joint latent representation as described previously or an image-only latent representation by training an image autoencoder independently of the graph autoencoder latent space. The image autoencoder used for computing the joint latent representation and the image autoencoder used for computing the image-only latent representation have the same architecture. An image patch size of $75.68 \times 75.68\ \mu m^2$ was used for training the autoencoders, which takes into account a neighborhood size comparable to the 20-nearest-neighbor cell adjacency used in the graph autoencoder. All regression models were trained with cells in: (1) cluster 1; (2) cluster 3; or (3) both cluster 1 and cluster 3. For training the regression model, we either used both 13-month samples or only the 13-month AD sample without the control. Descriptions for all models together with the corresponding classification errors can be found in Supplementary Data 1.

Plaque images were preprocessed by setting an intensity threshold of 10 to filter out noise, applying Gaussian blur with sigma of 10, a second intensity threshold of 100, and a minimum size filter of 1111 pixels ($8.95\ \mu m^2$). A cell was labeled as positive if there was plaque within the $75.68 \times 75.68\ \mu m^2$ image patch centered at the cell and it was labeled as negative otherwise. The hidden layers used leaky ReLU activation[69] and a dropout rate of 0.5. The output layer used ReLU activation because plaque size is non-negative. The output is a positive prediction if the output value is larger than 1111 pixels. 15% of randomly selected cells in each training tissue sample were held out for validation and testing. We chose the training epoch that resulted in approximately equal true positive rate (TPR) and true negative rate (TNR) based on the validation set (Supplementary Fig. 18b). If there was no epoch that resulted in both TPR and TNR <0.5, we concluded that the regression model was unable to train on the input data, i.e. the positive and negative inputs were indistinguishable (see Supplementary Data 1).

Keeping all other conditions the same, training the regression model on the joint latent space of cells consistently achieved lower classification error than training on the image-only latent space of the CNN autoencoder. This indicates that chromatin images contain additional information not reflected in gene expression. When only cluster 3 in the 13-months AD sample was used in training, the regression model was unable to train, i.e., there was no epoch that resulted in both TPR and TNR <0.5. The regression model was able to train after adding more negative cells by incorporating the 13-months control sample in training. This is another indication that cluster 3 is more advanced in AD progression and that cells further away from plaque in cluster 3 have similar gene expression, cell neighborhood,

and nuclear morphology and condensation patterns as cells close to plaque (within the input image patch).

All downstream analyses involving regression gradients were performed using the regression model trained on the joint representation of the cells in cluster 1 and 3 in both 13-month samples (Supplementary Data 1, model #14). The gradient of plaque size with respect to the input chromatin images was calculated using backpropagation and activation mapping. We backpropagated the gradients from the regression output of plaque size to the joint latent space. This activation of latent features was further backpropagated through the CNN encoder to the last convolutional layer to calculate an average weight for each channel in the last layer. These weights were used to calculate a weighted average of the channels in the last convolutional layer which was then projected to the input DNA image. This activation mapping from the latent features to the input image is adapted from the Grad-CAM method[54]. The 3D segmentation mask was projected to the x-y plane to obtain the average gradient within each cell.

### Chromatin condensation

Chromatin pixel intensities of each cell, used for calculating the heterochromatin ratio, were obtained from the 3D chromatin images of cells after 3D segmentation. The histograms of chromatin pixel intensities were normalized by the total number of chromatin pixels of each cell type in the given tissue region, i.e. the histogram bins of each sample sum to 1 (Figs. 4e, 5a, Supplementary Figs. 24–26). Chromatin pixels within a cell were divided into either euchromatin pixels or heterochromatin pixels based on intensities. The threshold for identifying heterochromatin pixels was calculated for each cell type at each time point using all cells in the cortex of the control sample at the given time point. Following prior work[67], this threshold was calculated as $(0.4 \times max + min + 0.35 \times (max-min))/2$, where max and min are the maximum and minimum of all pixel intensities of a given cell type in the control sample. The heterochromatin ratio of a cell was then defined as the total chromatin pixel intensity of heterochromatin in the cell normalized by the total chromatin pixel intensity of the cell. When fitting the linear regression between heterochromatin ratio and regression gradient, we removed outlier cells with $\log_2$ average gradient <4 (Fig. 4g, Supplementary Fig. 23). For plotting the pie charts by grouping cells based on their heterochromatin ratio and gradient, we used a threshold of 6.5 for the $\log_2$ gradient for excitatory neurons and 6.7 for glial cells in order to balance the cell numbers, because we found glial cells to be more predictive of nearby plaque (Supplementary Fig. 21). For the threshold for heterochromatin ratio we used 0.5 for excitatory neurons and 0.8 for glial cells in order to balance the cell numbers, because we found glial cells to have more condensed nuclei in general (Supplementary Fig. 22).

### Differential expression analysis and gene ontology enrichment

The differential expression analysis was performed with the SCANPY package[24]. Statistical significance was defined as $p$-value < 0.05 after correction using the Benjamini-Hochberg procedure[73] and fold change of at least 10% in either direction. Gene ontology and enrichment analysis were performed using the Gene Ontology database (release date 2021-12-15)[74–76]. The list of all 2112 genes in the STARmap PLUS gene expression matrix was used as the background for the enrichment analysis.

### Statistics and reproducibility

Eight mouse brain samples were used for the STARmap PLUS experiment as described in Zeng et al.[4]. No data published in the STARmap PLUS paper was excluded from our analyses. The initial four mouse samples (8-months AD and control, 13-months AD and control) were replicated with four additional mice samples. Our analyses on the replicates reproduced the results obtained on the initial four mice

samples (Supplementary Fig. 2). The experiments were not randomized. The Investigators were not blinded to allocation during experiments and outcome assessment. The mice used for STARmap PLUS include the pR5-183 line expressing the P301L mutant of human tau and PS2$_{N141I}$ and APP$_{swe}$ (PS2APP$^{homo}$; P301L$^{hemi}$) and non-transgenic control. Mouse brains were transferred to cryostat (Leica CM1950) for tissue sectioning. The STARmap PLUS protocol[4] was then applied to the tissue sections to obtain the spatial RNA and protein signals. Image processing was implemented in MATLAB R2019b, which includes multi-dimensional histogram matching, tophat filtering, image registration, spot calling, barcode filtering, and 2D cell segmentation[4].

### Reporting summary

Further information on research design is available in the Nature Portfolio Reporting Summary linked to this article.

## Data availability

All datasets used in this work are publicly available from the following sources: The STARmap PLUS data was obtained from Zeng et al.[4] and is available at https://singlecell.broadinstitute.org/single_cell/study/SCP1375/integrative-in-situ-mapping-of-single-cell-transcriptional-states-and-tissue-histopathology-in-an-alzheimer-disease-model as well as on Zenodo at: https://doi.org/10.5281/zenodo.7332091[77]. The reference gene expression used for cell type classification of the predicted gene expression is available from the Allen Brain Map[53]: https://portal.brain-map.org/atlases-and-data/rnaseq/mouse-whole-cortex-and-hippocampus-10x. The 10x Visium dataset is available at https://www.10xgenomics.com/resources/datasets/multiomic-integration-neuroscience-application-note-visium-for-ffpe-plus-immunofluorescence-alzheimers-disease-mouse-model-brain-coronal-sections-from-one-hemisphere-over-a-time-course-1-standard. A Source Data file is provided and contains statistics of differential expression reported in Fig. 5b and Supplementary Fig. 29, including p-values and log fold changes. Source data are provided with this paper.

## Code availability

Our code and a list of required open-source software packages are available at https://github.com/uhlerlab/STACI and on Zenodo: https://zenodo.org/record/7300119#.Y2k0VS-B35g[78].

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

## Acknowledgements

X.Z. was supported by the Eric and Wendy Schmidt Center at the Broad Institute. X.W. was supported by a Searle Scholarship, Cabot Professorship at MIT, Edward Scolnick Professorship at the Broad Institute, Merkin Institute Fellowship, Ono Pharma Breakthrough Science Initiative Award, and an NIH New Innovator Award (DP2 GM146245-01). G.V.S. was supported by ETH funding. C.U. was supported by NSF (DMS-1651995), ONR (N00014-17-1-2147 and N00014-22-1-2116), the MIT-IBM Watson AI Lab, MIT J-Clinic for Machine Learning and Health, and a Simons Investigator Award. The Titan Xp used for this research was donated by the NVIDIA Corporation.

## Author contributions

All authors designed the research. X.Z. developed and implemented the algorithms and performed model and data analysis. All authors wrote the paper.

## Competing interests

X.W. is an inventor of IP applications of STARmap and STARmap PLUS and a scientific co-founder of Stellaromics. The other authors declare no competing interests.
