## [Peer Review File · Nature Communications]

Reviewers' Comments:

Reviewer #1:

Remarks to the Author:

The authors propose a method for jointly analyzing scRNAseq data with spatial location information based on an autoencoder that uses a shared latent space across the multiple modalities. The graph-based autoencoder utilizes a separate decoder for each of the modalities considered: gene expression, cell location, and chromatin images.

- The ability to train on both chromatin images and transcriptomic profiles but then sample from only chromatin images is very interesting. The use of both a reconstruction loss and a divergence from the transcriptomic latent expression is well motivated and appears successful. The held-out experiment validates the approach.
- The graph-based autoencoder that encodes and decodes each node seems well suited to this integration task and is strongly motivated and demonstrated.
- The use of the gradient of the image encoder as a saliency measure is an interesting way of showing the areas of the image are important for classification. The quantitative analysis possible from this was especially strong.
- The use of the spatial information as a constant factor that connects the different samples and thus mitigates batch effects is clever and demonstrated to be of significant help. It is surprising that no additional batch effect mitigation techniques are needed in this context.
- The only description of the clustering method that is being used is relegated briefly to the end. Given how much discussion of the clusters themselves there is, this seems important enough to be given more attention (in the main paper). Especially given the motivation of other choices (e.g. the larger latent layer) is justified by the quality of these clusters, the specific clustering method is important. It would also be stronger to use multiple clustering methods and show the conclusions about these design choices hold.
- Why is the graph autoencoder trained and fixed prior and separately to the CNN autoencoder? Was the alternative explored, where they were trained simultaneously and jointly? It could be imagined that the latent structure could be morphed such that it is easier for the CNN latent structures to be accommodated.
- The discussion of the size of the cell neighborhood feels incomplete. In the first sentence ("Reducing the size of a cell's neighborhood by half... reveals temporal dynamics"), it is unclear what the original neighborhood size is and how the relative halving relates to a temporal axis in any way. Also, while it is briefly mentioned or alluded to in a few places, there should be a more explicit explanation of how the cell neighborhoods are defined (what distances are used for the knn construction, etc...)
- It would be nice to see a comparison to other methods in the field, for example integrated diffusion which also learns a joint embedding from two modality graphs (<https://ieeexplore.ieee.org/document/9596214>).

Reviewer #2:

Remarks to the Author:

In this article, the authors introduce a new computational software to integrate spatial transcriptomics data with chromatin imaging data using graph-based autoencoders. The authors claim that STACI can represent multiple datasets in a single-joint representation, which is very helpful to interpret orthogonal datasets in biological context. The authors then use unpublished spatial transcriptomics data from another paper (Zheng et al., bioRxiv) in TauAPP mouse model of AD to jointly understand gene-expression, chromatin compactness in the same data. The algorithm looks good, but a) is not available online in github (link not working) and b) does not leverage snRNA and snATAC-seq datasets from same samples in the same joint representation. Some of the major comments are listed -

1. The STARmap plus data also has information about proteins identified using primary antibodies. It is unclear if STACI can integrate protein expression data from STARmap as well as other multiplexed datasets.

2. The robustness of the method should not be limited to unpublished datasets, rather should be expanded to published datasets like Chen et al. All of the analysis is dependent on the quality of the STARmap plus data which has not been peer-reviewed. Moreover, STARmap plus is not a commercially available method, limiting its usefulness to labs which have developed it.
3. The methods section is not described properly for a methods paper. For example no information is provided how chromatin imaging data were obtained and processed.
4. Does STACI need a paired format to analyze chromatin imaging and spatial transcriptomics data from the same samples. What is the accuracy/robustness of the method if different biological samples are used for different data modalities? Can the authors simulate and quantify this accuracy?
5. One of the biggest limitations for review is that the STACI github link is not online, presumably as the code is not ready. It becomes extremely difficult to understand how the pipeline works and if it can be readily adapted. It seems that no one outside of the authors have used this code, as it is not available which is a big negative sign.
6. Can the STACI pipeline be scaled to large datasets consisting of tens of samples from spatial transcriptomics data? A computational resource (CPU/GPU/RAM) usage plot will be very helpful to understand how the pipeline will behave in large datasets.
7. It is unclear how good the performance will be if using other spatial modalities like MERSCOPE and 10x genomics visium. Several available datasets are available, and the authors must show that the pipeline works with minimal changes from the user. A tutorial of such varied datasets should also be included in the github page.
8. How does STACI pipeline differ from the previously published work from the same group - cross-modal autoencoders (Yang et al., Nat Comm, 2021). Can STACI be used to also integrate single-cell RNA and ATAC-seq data with spatial transcriptomics data?
9. Can STACI be used to deconvolute cell-type composition in spot-based spatial transcriptomics datasets like 10x genomics visium platform. In theory, it should, but the authors need to explore it further.

Reviewer #3:

Remarks to the Author:

This work introduced a graph neural network model to combine gene expression, cell position and image modalities into a joint representation using the STARmap datasets. Authors provide a number of results but I am not sure they are evaluated in a proper way.

[Biological task]

There are a number of publications doing similar tasks but authors did not mention nor compare in the manuscript.

- For combining gene and spatial position:

Stlearn: <https://www.biorxiv.org/content/10.1101/2020.05.31.125658v1>

HMRf: <https://www.nature.com/articles/nbt.4260>

- For combining image and gene:

MUSE: <https://www.nature.com/articles/s41587-022-01251-z>

Stnet: <https://ai.stanford.edu/~bryanhe/publications/stnet.pdf>

I believe there are also a number of other publications in this field.

- With that, some claims in the introduction are not accurate, such as "Technologies such as STARmap and 10x Genomics Visium measure chromatin staining paired with spatial transcriptomics, but current analysis methods do not make use of this information."

- It would be valuable to benchmark the proposed model with existing works using datasets with

"ground truth" to show the benefits.

[Organization]

I have a major concern on this point.

- The manuscript is not in a good organization. Figures are arranged in an inconsistent way with the main text (especially for Figures 2-3 and ED figures). For the same result section, readers need to jump to different figures to see the related results. I think it might not be a good way to have one storyline in context but provide another different storyline in the figure organization.
- The description of methods is divided to several sections, which makes it hard to understand what conditions/frameworks were used to obtain the results. E.g. Fig 2b comes before introducing overparameterization and ZINB, but it is unclear this result was obtained with or without using these two elements.

[Machine learning model]

It is really good to discuss the benefits from each architecture design.

- How to balance the reconstruction errors from each modalities. I think the reconstruction from image is quite different from expressions or positions.
- Current comparisons are among variants from the same GNN framework. Finding an optimal framework under GNN cannot guarantee a comparable performance comparing with existing tools in the field.
- Experiment conditions: due to the organization, it is unclear what settings were used for each experiment.
- Evaluations: (as described below), current metrics are not objective and cannot be viewed as ground truth.
- As authors said, it is unclear how overparameterization works. I think it is risky to make hypothesis that it can reduce batch bias and emphasize biological signals, as it has no information about the difference between batch and true signal.

[Data analysis]

- For image feature learning, as I can tell from STARmap technology, you can actually extract exact single cell images using segmentation masks. Will this be more accurate than the squared segmentation?

[Experiments]

Batch effects:

- In Fig 2g, h, I don't think there is a strong link between these 3 clusters and plaques, as you also identified three similar clusters from control mice. I suspect this is a consequence from different subregions. Another way to check is to perform differential expression analysis among these regions and see what comes out.
- Following the above point, there is no clear evidence that these 3 regions are meaningful in biology. Given the small number of plaques, the statistical test is not reliable. I think it is less reasonable to compare results based on these 3 clusters.
- ED Fig 3: spatial coherence of clusters cannot be used as a ground truth without additional evidence -- as we know some cell types are distributed randomly in brain.
- In ED Fig 4, as 4 tissue slices are from different biological conditions, we do not 100% sure they should be overlapped completely -- additional evidence is needed.
- To test the model on batch bias removal, authors can use multiple scRNAseq data sets with batch labels and check how the overparameterization works to remove the bias.
- There are a number of approaches handling the batch bias problem, can you compare with them?
- Visualization is not a good measurement of batch bias. You might want to use quantitative metric to measure them, such as entropy of mixing

Gene prediction:

- Fig 3a, predicting the expression of 8 month control sample using models trained on other samples -- will biological differences affect the result?

Plaque size prediction:

- I think this section did not illustrate the benefits from the model. If most informative features are from image, will image alone analysis are more meaningful instead of a combined analysis?

[Tool]

- Link provided is invalid.

Reviewer #1 Comments and Responses:

The authors propose a method for jointly analyzing scRNAseq data with spatial location information based on an autoencoder that uses a shared latent space across the multiple modalities. The graph-based autoencoder utilizes a separate decoder for each of the modalities considered: gene expression, cell location, and chromatin images.

-- The ability to train on both chromatin images and transcriptomic profiles but then sample from only chromatin images is very interesting. The use of both a reconstruction loss and a divergence from the transcriptomic latent expression is well motivated and appears successful. The held-out experiment validates the approach.

-- The graph-based autoencoder that encodes and decodes each node seems well suited to this integration task and is strongly motivated and demonstrated.

-- The use of the gradient of the image encoder as a saliency measure is an interesting way of showing the areas of the image are important for classification. The quantitative analysis possible from this was especially strong.

-- The use of the spatial information as a constant factor that connects the different samples and thus mitigates batch effects is clever and demonstrated to be of significant help. It is surprising that no additional batch effect mitigation techniques are needed in this context.

We thank the reviewer for these positive comments.

-- The only description of the clustering method that is being used is relegated briefly to the end. Given how much discussion of the clusters themselves there is, this seems important enough to be given more attention (in the main paper). Especially given the motivation of other choices (e.g. the larger latent layer) is justified by the quality of these clusters, the specific clustering method is important. It would also be stronger to use multiple clustering methods and show the conclusions about these design choices hold.

We thank the reviewer for this suggestion. A description of the clustering method and the corresponding parameters used to obtain the results in Figure 2 in the main text was added to section 2 of the Results in the main paper: "All clusters were obtained by applying Leiden clustering to the top 40 principal components of the latent representation with a clustering resolution of 0.1."

We also performed a careful analysis of the effect of using other parameters in Leiden clustering as well as other clustering methods. The results are shown below in Fig. 1 and 2 (which were added as Extended Data Fig. 7 and 8 in the revised manuscript) and demonstrate that the same conclusions about design choices (e.g. the larger latent layer) are obtained using different parameters of Leiden clustering or using k-means clustering or hierarchical clustering. With the same over-parameterized model as in Figure 2b in the main text, we used a lower resolution of 0.05 to obtain less clusters (Fig. 1b below) and a higher resolution of 0.18 to obtain more clusters (Fig. 1c below), both of which show that clusters in the latent representation correspond to the same tissue region across different samples. In contrast, if we use the standard autoencoder with an under-parameterized latent layer as in Figure 2c in the main text, both Leiden clustering resolutions at 0.05 and 0.18 show sample-to-sample variations (see Fig. 1d below). For example, the cortex region in the two 13-months samples cannot be assigned to the same cluster. Similarly, clustering the latent representations of the over-parameterized model using k-means clustering or hierarchical clustering results in consistent separation of all samples into different cortex regions, CA, and DG (see Fig. 1e, 2a, 2c, 2d below), whereas k-means clustering of the under-parameterized model results in some cells from the cortex, CA, and DG being grouped into the same cluster (see red dots in Fig. 1f below); it is also apparent that the segmentation of known anatomical regions is worse than using k-means clustering with the over-parameterized model. For hierarchical clustering, the over-parameterized model also has more consistent segmentation of tissue regions across samples and more consistent segmentation with the known anatomical regions than the under-parameterized model (see Fig. 2

below). Hierarchical clustering of the under-parameterized model results in some cells of the cortex and DG in the same cluster (yellow dots in Fig. 2b below). For both k-means and hierarchical clustering, the number of clusters is set to 8 to be consistent with the number of clusters in Figure 2b of the main text.

The following sentences were added to section 2 of the Results in the main paper to describe these conclusions: “ The correction of batch effect using an over-parameterized latent space is also observed when different Leiden resolutions are used (Extended Data Fig. 7b-d) and when different clustering methods are applied (Extended Data Fig. 7e, 7f, 8).”

Fig. 1 (Extended Data Fig. 7b-f in the revised manuscript). Over-parameterization consistently removes batch effects with different clustering methods and parameters.

b) Leiden clustering of the latent space of our over-parameterized model with a latent dimension of 6000 and clustering resolution of 0.05. Each dot is a cell plotted with its physical coordinates in the tissue and colored by the cluster identity. 20 nearest neighbor graph is used as the adjacency.

c) Leiden clustering of the same model as in b) with a clustering resolution of 0.18.

d) Cells from the same cluster (blue) using our model without over-parameterization correspond to different regions in 13-month control (left) and 13-month AD (right). The latent dimension of this model is 1024. The results of two different clustering resolutions are shown.

- e) k-means clustering of the over-parameterized model in b) also results in consistent separation of all samples into different cortex regions, CA, and DG.
- f) k-means clustering of the under-parameterized model in d) results in some cells from the cortex, CA, and DG being grouped into the same cluster.

Fig. 2 (Extended Data Fig. 8 in the revised manuscript). Overparameterization consistently removes batch effects using hierarchical clustering with different linkages.

a) Hierarchical clustering of the latent space of our over-parameterized model with a latent dimension of 6000 and ward linkage. Each dot is a cell plotted with its physical coordinates in the tissue and colored by the cluster identity. 20 nearest neighbor graph is used as the adjacency.

b) Hierarchical clustering of the latent space of our under-parameterized model with a latent dimension of 1024 and ward linkage. Each dot is a cell plotted with its physical coordinates in the tissue and colored by the cluster identity. 20 nearest neighbor graph is used as the adjacency. The under-parameterized model is less consistent with the known anatomical regions and results in some cells of the cortex and DG in the same cluster.

c) Hierarchical clustering of the same model as in a) with complete linkage.

d) Hierarchical clustering of the same model as in a) with average linkage.

-- Why is the graph autoencoder trained and fixed prior and separately to the CNN autoencoder? Was the alternative explored, where they were trained simultaneously and jointly? It could be imagined that the latent structure could be morphed such that it is easier for the CNN latent structures to be accommodated.

We thank the reviewer for this question. The graph autoencoder is trained and fixed first because the gene expression matrix provides segmentation of single cells in terms of mRNA molecules, which allows the latent space to be computed with single-cell resolution. Thus, without additional image segmentation of the chromatin images, training and fixing the CNN autoencoder is not an alternative that would result in a latent space with the same resolution. Training simultaneously or jointly, another alternative suggested by the reviewer, could retain single-cell resolution from the gene expression matrix. However, as shown in a previous study by Yang and Uhler¹, training two autoencoders simultaneously increases computational costs but does generally not improve the performance of domain translation on a comparable task as predicting gene expression from images in our application (section 4.2, Figure 6 in Yang and Uhler 2019). In line with these prior observations, we observed that when training and fixing the graph autoencoder prior to the CNN autoencoder, the CNN autoencoder results in good reconstruction quality of the images (Extended Data Fig. 14). To further confirm this, we added Fig. 3 (which is Extended Data Fig. 15 in the manuscript) to show that the image reconstruction quality is comparable to training a CNN autoencoder alone without matching the latent space to the graph autoencoder. Thus, it is not necessary to update the graph autoencoder when training the CNN autoencoder to accommodate the CNN latent structure.

While this analysis shows that there are no disadvantages to fixing one autoencoder prior to the training of the other autoencoder, there are various benefits to doing so, including more stable training and less memory required for training, which is especially important in our over-parameterized model that has a larger memory requirement than the standard autoencoder. In addition, if there are additional modalities available to be incorporated into the joint latent space, the current setup where one autoencoder is fixed first does not require retraining of all the previously trained autoencoders.

Fig. 3 (Extended Data Fig. 15 in revised manuscript). Examples of input chromatin images to the CNN autoencoder and the corresponding reconstructed images. There is no latent space matching with the graph autoencoder. All samples were used in training the CNN autoencoder.

-- The discussion of the size of the cell neighborhood feels incomplete. In the first sentence ("Reducing the size of a cell's neighborhood by half... reveals temporal dynamics"), it is unclear what the original neighborhood size is and how the relative halving relates to a temporal axis in any way. Also, while it is briefly mentioned or alluded to in a few places, there should be a more explicit explanation of how the cell neighborhoods are defined (what distances are used for the knn construction, etc...)

We thank the reviewer for this comment. The original neighborhood size used to obtain Figure 2 is 20 nearest neighbors measured in Euclidean distance, which is reduced to 10 for the subsequent analysis. We clarified this by adding the following sentences to Results Section 1: "The definition of neighborhood of cells can be customized to the particular application. To demonstrate our model, we used a 20-nearest-neighbor adjacency matrix based on the Euclidean distance between the centroid of each cell in each sample to obtain the results in Figure 2. Reducing the size of a cell's neighborhood from 20 nearest neighbors by half ... reveals temporal dynamics".

There could be multiple types of spatial pattern simultaneously and, depending on the size of the neighborhood being investigated, some spatial patterns could dominate over others. The comparison of 10 vs 20 nearest neighbors showed that temporal difference dominates at a smaller neighborhood size. Due to the contribution from more distant neighbors at a larger neighborhood size, perhaps other differences start to dominate, e.g. difference in anatomical regions, while the temporal difference still exists. Extended Data Fig. 6b showed that temporal difference is still a strong factor in the DG region and the bifurcating trajectory obtained using 10 nearest neighbors can be recovered by subclustering the cluster corresponding to the DG region.

-- It would be nice to see a comparison to other methods in the field, for example integrated diffusion which also learns a joint embedding from two modality graphs (<https://ieeexplore.ieee.org/document/9596214>).

We appreciate the related work pointed out by Reviewer 1. In the following, we provide a careful comparison between the integrated diffusion method and our method; we also added a paragraph in the introduction to thoroughly discuss related methods (including the integrated diffusion method as well as other related methods mentioned by reviewer 3) and how they differ from STACI.

The integrated diffusion method learns a joint embedding by combining the modality specific diffusion operators of the two modalities. As such, the integrated diffusion method does not provide a method for incorporating spatial coordinates. One could for example consider using Euclidean coordinates with a Gaussian kernel in order to compute local affinity. But it is not clear a-priori that this would lead to a meaningful diffusion operator. We developed STACI to obtain a joint embedding of three modalities, i.e. gene expression, spatial coordinates, and chromatin images, but it is not clear how to do this using integrated diffusion. By training additional autoencoders and matching the latent space of the new autoencoder to the existing joint latent space by l_2 loss, our framework could easily accommodate also additional modalities. Another important feature of STACI that is in contrast to the integrated diffusion method is that it can be used to predict one of the modalities given another using the joint latent space (as demonstrated in section 3 of the Results to translate from chromatin imaging to transcriptomic modality). Finally, since the experiments of the integrated diffusion paper are focused on denoising of the datasets by removing added Gaussian noise for each of the two modalities, it is not clear whether it could remove systematic bias between samples within each modality, such as batch effects, and it is also unclear how batch effects might aggregate when two modalities are combined through the integrated diffusion operator. By using over-parameterization, STACI identifies a latent space that emphasizes the major direction of variation, thereby reducing batch effects, as we demonstrated in Results Section 2.

We carefully updated the main text to describe various related works and how they differ from our method: "With the rise of spatial transcriptomic data, several computational approaches have been developed to integrate different data modalities in the tissue context. STACI is the first method, to our knowledge, that simultaneously integrates all the available modalities, namely gene expression, cellular neighborhoods, and chromatin imaging, and is capable of translating between different data

modalities and identifying combined morphometric and molecular disease biomarkers in the tissue context. In particular, various methods have been developed that integrate or can be adapted to integrate single-cell gene expression with images, but do not incorporate cell location into their analysis.³²⁻³⁴ In addition, other methods such as HMRF³⁵ incorporate spatial information into the analysis of gene expression to identifying spatial regions with consistent patterns of cell states but do not make use of imaging data. It is the joint latent representation of gene expression and cell location used by STACI that enables us to incorporate chromatin imaging data into the analysis and perform various downstream analysis, such as clustering cells into finer spatial regions, without retraining the model. An interesting method that integrates all three modalities available in spatial transcriptomic datasets is stLearn, a denoising approach that replaces the gene expression of a cell by the average expression of its neighboring cells, weighted by their image similarity.³⁰ In contrast, STACI aims to identify patterns in cell neighborhoods, which can consist of diverse cell types/states, by taking into account all cells in the physical neighborhood regardless of morphological similarity. Both stLearn and HMRF are unable to predict gene expression from images or identify morphological disease markers associated with the identified tissue regions. Finally, another key feature of STACI compared to current methods for multimodal integration is the built-in correction of batch effects that applies to all modalities in the joint latent space.”

Reviewer #2 Comments and Responses:

In this article, the authors introduce a new computational software to integrate spatial transcriptomics data with chromatin imaging data using graph-based autoencoders. The authors claim that STACI can represent multiple datasets in a single-joint representation, which is very helpful to interpret orthogonal datasets in biological context. The authors then use unpublished spatial transcriptomics data from another paper (Zheng et al., biorXiv) in TauAPP mouse model of AD to jointly understand gene-expression, chromatin compactness in the same data. The algorithm looks good, but a) is not available online in github (link not working) and b) does not leverage snRNA and snATAC-seq datasets from same samples in the same joint representation. Some of the major comments are listed -

We thank the reviewer for the positive and helpful comments.

- a) In the initial submission we had provided all code related to this manuscript in a zip file. We thank the reviewer for pointing out the GitHub link issue. The repository is now public.
- b) We agree with the reviewer that one should integrate all available modalities in the same joint representation, which is what our model does. For the STARmap dataset analyzed in our paper, there is no snRNA-seq or snATAC-seq data available on the same samples. When additional modalities (such as snRNA-seq or snATAC-seq) are available, our framework can easily integrate these by training an additional autoencoder per new modality and matching the latent space of the new autoencoder with the previously trained joint latent representation. The integration of unpaired single-cell datasets can be achieved by changing the l_2 loss used for matching the latent space of the two autoencoders to KL divergence, as demonstrated in earlier work from our group (Yang et al., Nat Comm, 2021²). It is interesting to note that when a new modality becomes available, it is not necessary to retrain the previously trained autoencoders. One can use the same procedure as we describe in our manuscript for integrating the chromatin imaging data, namely training the CNN autoencoder for images to match its latent space to the latent space of the graph autoencoder, without updating the graph autoencoder. We added a sentence to Results Section 3 to clarify how other data modalities (if available) could be integrated into the joint latent space: "Also unpaired datasets (such as scRNA-seq or scATAC-seq, if available) can be integrated without updating the existing autoencoders by training an additional autoencoder per new modality and choosing an appropriate distance metric, such as KL divergence³⁰, for matching the latent representation of the new modality with the existing joint latent representation."

Below, we provide a point-by-point response to each comment.

1. The STARmap plus data also has information about proteins identified using primary antibodies. It is unclear if STACI can integrate protein expression data from STARmap as well as other multiplexed datasets.

STACI can easily incorporate protein expression data or other multiplexed datasets into the joint latent space by using multi-channel cell images as input to the CNN autoencoder, where each channel represents either a protein image or the chromatin image. The CNN autoencoder can be adapted by changing the first convolutional layer to have the same number of channels as the input images. We did not use the protein images to train our model because we intended to hold out the protein images as a validation for the analysis of the latent space clustering of STACI. For example, we used plaque as an orthogonal measurement to validate that the three clusters in the cortex identified by STACI are biologically meaningful regions which have different amounts of plaque accumulation in AD. We added a sentence to Results Section 3 to clarify how protein expression data or other multiplexed datasets (if available) could be integrated into the joint latent space: "We here demonstrate the integration of images with only the chromatin channel to hold out the plaque images as an orthogonal validation of our analysis, but plaque and other multiplexed imaging channels (if available) can be incorporated into the joint latent space by using multi-channel cell images as input to the CNN autoencoder."

2. The robustness of the method should not be limited to unpublished datasets, rather should be expanded to published datasets like Chen.et al. All of the analysis is dependent on the quality of the STARmap plus data which has not been peer-reviewed. Moreover, STARmap plus is not a commercially available method, limiting its usefulness to labs which have developed it.

We thank the reviewer for this comment. Because 10x Visium datasets have been widely used for benchmarking and the technology is commercially available, in order to demonstrate the robustness of our method, we applied our method to a 10x Visium dataset available from the 10x website

(<https://www.10xgenomics.com/resources/datasets/multiomic-integration-neuroscience-application-note-visium-for-ffpe-plus-immunofluorescence-alzheimers-disease-mouse-model-brain-coronal-sections-from-one-hemisphere-over-a-time-course-1-standard>). The results are shown below in Figures 4, 5 and 6, which we added into our manuscript as Extended Data Fig. 10-12. We also added the following sentences to Results Section 2 describing the results we obtain using STACI on this benchmark dataset: "Our approach is also applicable to other spatial transcriptomics technologies, such as the commercially available 10x Visium platform.... To demonstrate this, we applied STACI to a 10x Visium dataset of 12 mouse brain samples consisting of AD and control mice at different time points⁵². Compared to the clusters given by 10x based on gene expression alone (Extended Data Fig. 10a, 11), STACI achieves more consistent results across all samples, given the known anatomical regions of mouse brains (Extended Data Fig. 10c, 12). The computational resources required for this dataset were recorded over six training epochs (Extended Data Fig. 13)."

On a separate note, the STARmap/plus method is under active commercialization by Stellaromics.

Fig. 4 (Extended Data Fig. 10 in revised manuscript). Validation of STACI on 10x Visium dataset of 12 mouse brain coronal sections.

a) Clustering of gene expression provided on the 10x website².

b) Leiden clustering of STACI latent space without using cell adjacency as input and with a latent dimension of 30000. The clustering resolution is 0.4.

c) Leiden clustering of the full STACI model with a latent dimension of 30000. The clustering resolution is 0.4. This clustering result shows consistent tissue segmentation across all samples and is consistent with known anatomical regions of mouse brains. In contrast, other existing methods for the analysis of spatial transcriptomics data result in having spots from different anatomical regions in the same clusters. For example, the tutorial of stLearn³ also shows an application to 10x Visium data of mouse brain coronal sections (https://stlearn.readthedocs.io/en/latest/tutorials/stSME_clustering.html), but one cluster contains cells from the hippocampus region, isocortex, olfactory areas, and cortical subplate (cluster 18) and another cluster contains spots from both hypothalamus and cerebral nuclei (cluster 4). The Giotto package's implementation of HMRF⁴ also demonstrates the results on 10x Visium data of mouse brain coronal sections (https://rubd.github.io/Giotto_site/articles/mouse_visium_brain_201226.html). While the accuracy and resolution of the spatial domains identified by HMRF and STACI are mostly comparable, some HMRF domains contain spots from different known anatomical regions of the brain in the same domains; for example, domain 16 contains cells from both inside and outside of the hypothalamus. Similarly, MUSE⁵, another existing method for the analysis of spatial transcriptomics data, shows mixing of cells from different brain regions into the same cluster (see Figure 6e⁵); for example, clusters 1, 4, and 15 show mixing of cells from the isocortex (CTX), hypothalamus (HY), and olfactory area (OLF), and cluster 0 contains a mixture of cells from the hippocampus and various other regions.

Clustering provided by 10x

Fig. 5 (Extended Data Fig. 11 in revised manuscript). Segmentation of the hippocampus regions according to the clustering of gene expression provided on the 10x website.

Cells in cluster 5 or cluster 14 are colored blue in the respective plots.

Fig. 6 (Extended Data Fig. 12 in revised manuscript). Segmentation of the hippocampus regions according to the clustering in the latent space of the full STACI model with a latent dimension of 30000.

Cells in cluster 11 or cluster 3 are colored blue in the respective plots.

3. The methods section is not described properly for a methods paper. For example no information is provided how chromatin imaging data were obtained and processed.

We thank the reviewer for this comment. Based on the reviewer's comment, we added the following paragraph to the Methods Section to provide more information on the experimental setup of how the chromatin images were obtained: "Experimental setup for imaging: All images were obtained with a Leica TCS SP8 confocal microscope and with a 40x objective. The voxel size is $0.0946 \times 0.0946 \times 0.3463 \mu\text{m}^3$ (x, y, and z dimensions respectively). Propidium Iodide (PI) staining was applied according to the manufacturer's protocol. A detailed protocol of the experiment can be found in the STARmap PLUS paper²."

With respect to processing of the chromatin imaging data, we feel we had carefully described this in the Methods Section. For example, the 3D segmentation of the chromatin images was carefully described in "3D segmentation of chromatin images" in the Methods section: "The python package py-clesperanto of CLIJ was used for the 3D segmentation of chromatin images³. For each cell, we used all the z-stack images and cropped the horizontal directions to $37.84 \times 37.84 \mu\text{m}^2$ centered at the cell centroid. After min-max scaling to $[0, 1]$, the 3D stack of each cell was further cropped to $18.92 \times 18.92 \mu\text{m}^2$ in the horizontal directions. The images were resampled in the z-direction to have isotropic voxels. Then we applied Gaussian blur, spot detection, a second Gaussian blur with sigma set to 3, Otsu thresholding, and Voronoi labeling. The first Gaussian blur was optimized with two iterative searches to obtain the maximum sigma value for which a cell can be detected at the given centroid. We used "binary_fill_holes" in the SciPy package with a $2 \times 2 \times 3$ matrix of ones as the structuring element on the resulting mask after Voronoi labeling." In addition, the procedure for calculating the heterochromatin ratio of each cell using the chromatin pixel intensities in the cell was carefully described in "Chromatin condensation" in the Methods section: "The threshold for identifying heterochromatin pixels was calculated for each cell type at each time point using all cells in the cortex of the control sample at the given time point. Following prior work, this threshold was calculated as $(0.4 \times \text{max} + \text{min} + 0.35 \times (\text{max} - \text{min}))/2$, where max and min are the maximum and minimum of all pixel intensities of a given cell type in the control sample. The heterochromatin ratio of a cell was then defined as the total chromatin pixel intensity of heterochromatin in the cell normalized by the total chromatin pixel intensity of the cell."

4. Does STACI need a paired format to analyze chromatin imaging and spatial transcriptomics data from the same samples. What is the accuracy/robustness of the method if different biological samples are used for different data modalities? Can the authors simulate and quantify this accuracy?

With the current setup described in the paper, a paired format is required, and usually paired images can be easily obtained for most spatial transcriptomics methods, including commercially available methods, e.g. MERFISH, seqFISH, and 10x visium. For example, such paired datasets have been obtained in various previous studies.⁴⁻⁷ As described above in the response to question 2 of the reviewer, we now also demonstrate the performance of STACI on a 10x Visium dataset.

If images can only be obtained for adjacent slides, such as in Slide-seq, then STACI can be adapted by changing the l_2 loss used for matching the latent space of the two autoencoders to KL divergence, as demonstrated in Yang et al., Nat Comm, 2021² and also described above in the response to the reviewer's first comment.

5. One of the biggest limitations for review is that the STACI github link is not online, presumably as the code is not ready. It becomes extremely difficult to understand how the pipeline works and if it can be readily adapted. It seems that no one outside of the authors have used this code, as it is not available which is a big negative sign.

In the initial submission we had provided all code related to this manuscript in a zip file. We thank the reviewer for pointing out the GitHub link issue. The repository is now public.

6. Can the STACI pipeline be scaled to large datasets consisting of tens of samples from spatial transcriptomics data? A computational resource (CPU/GPU/RAM) usage plot will be very helpful to understand how the pipeline will behave in large datasets.

STACI can easily be scaled to large datasets consisting of tens of samples of spatial transcriptomics data. To demonstrate this, we applied our method to a 10x Visium dataset available from the 10x website (<https://www.10xgenomics.com/resources/datasets/multiomic-integration-neuroscience-application-note-visium-for-ffpe-plus-immunofluorescence-alzheimers-disease-mouse-model-brain-coronal-sections-from-one-hemisphere-over-a-time-course-1-standard>). The result is demonstrated in Fig. 4-6 above (which is Extended Data Fig. 10-12 in the revised manuscript). We also added text to Results Section 2 describing the results we obtain using STACI on this benchmark dataset (see our response to point 2 above). This dataset has a total of 12 samples and 36,801 spots under tissue. We added a table summarizing the usage over 6 training epochs shown in Fig. 7 below (which is Extended Data Fig. 13 in the revised manuscript). The complete json trace file is added as a supplementary file and can be opened in Chrome trace viewer (<chrome://tracing>). The following sentence was added to Results Section 2 to describe this additional analysis: "The computational resources required for the analysis of this dataset were recorded over six training epochs (Extended Data Fig. 13 and supplementary file)."

Name	Self CPU %	Self CPU	CPU total %	CPU total	CPU time avg	Self CUDA	Self CUDA %	CUDA total	CUDA time avg	CPU Mem	Self CPU Mem	CUDA Mem	Self CUDA Mem	# of Calls
aten::mm	0.05%	17.132 ms	37.99%	12.595 s	74.971ms	16.370s	64.57%	16.648s	99.094ms	8 b	8 b	160.44 Gb	136.66 Gb	168
void cutlass::Kernel<cutlass_80_tensorop_s1 688gemm_1...	0.00%	0.000us	0.00%	0.000us	0.000us	4.312s	17.01%	4.312s	165.862 ms	0 b	0 b	0 b	0 b	26
void cutlass::Kernel<cutlass_80_tensorop_s1 688gemm_1...	0.00%	0.000us	0.00%	0.000us	0.000us	4.296s	16.95%	4.296s	143.210 ms	0 b	0 b	0 b	0 b	30
aten::addmm	0.05%	15.090 ms	53.30%	17.668 s	196.310 ms	4.085s	16.11%	4.443s	49.364ms	0 b	0 b	42.00 Gb	-9.95 Gb	90
void cutlass::Kernel<cutlass_80_tensorop_s1 688gemm_1...	0.00%	0.000us	0.00%	0.000us	0.000us	3.606s	14.23%	3.606s	78.402ms	0 b	0 b	0 b	0 b	46
aten::copy_	0.15%	48.478 ms	75.19%	24.925 s	27.151ms	3.202s	12.63%	3.202s	3.488ms	8 b	32 b	3.70 Gb	875.33 Mb	918
void cutlass::Kernel<cutlass_80_tensorop_s1 688gemm_1...	0.00%	0.000us	0.00%	0.000us	0.000us	2.780s	10.96%	2.780s	79.419ms	0 b	0 b	0 b	0 b	35
Memcpy PtoP (Device -> Device)	0.00%	0.000us	0.00%	0.000us	0.000us	2.293s	9.05%	2.293s	8.494ms	0 b	0 b	0 b	0 b	270
void cutlass::Kernel<cutlass_80_tensorop_s1 688gemm_1...	0.00%	0.000us	0.00%	0.000us	0.000us	2.219s	8.75%	2.219s	138.662 ms	0 b	0 b	0 b	0 b	16
void cutlass::Kernel<cutlass_80_tensorop_s1 688gemm_2...	0.00%	0.000us	0.00%	0.000us	0.000us	1.959s	7.73%	1.959s	78.340ms	0 b	0 b	0 b	0 b	25
Self CPU time total: 33.150s														
Self CUDA time total: 25.352s														
Name	Self CPU %	Self CPU	CPU total %	CPU total	CPU time avg	Self CUDA	Self CUDA %	CUDA total	CUDA time avg	CPU Mem	Self CPU Mem	CUDA Mem	Self CUDA Mem	# of Calls
aten::mm	0.08%	22.054 ms	37.22%	10.235 s	60.923ms	13.372s	63.20%	13.600s	80.952ms	0 b	0 b	153.07 Gb	130.38 Gb	168
void cutlass::Kernel<cutlass_80_tensorop_s1 688gemm_1...	0.00%	0.000us	0.00%	0.000us	0.000us	3.564s	16.84%	3.564s	118.791 ms	0 b	0 b	0 b	0 b	30
aten::addmm	0.07%	20.012 ms	52.46%	14.427 s	160.300 ms	3.443s	16.27%	3.736s	41.510ms	0 b	-8 b	33.62 Gb	-5.94 Gb	90
void cutlass::Kernel<cutlass_80_tensorop_s1 688gemm_1...	0.00%	0.000us	0.00%	0.000us	0.000us	3.207s	15.16%	3.207s	89.079ms	0 b	0 b	0 b	0 b	36
void cutlass::Kernel<cutlass_80_tensorop_s1 688gemm_1...	0.00%	0.000us	0.00%	0.000us	0.000us	2.897s	13.69%	2.897s	80.469ms	0 b	0 b	0 b	0 b	36
aten::copy_	0.20%	56.220 ms	74.44%	20.472 s	22.301ms	2.874s	13.58%	2.874s	3.131ms	16 b	8 b	7.20 Gb	7.24 Gb	918
void cutlass::Kernel<cutlass_80_tensorop_s1 688gemm_1...	0.00%	0.000us	0.00%	0.000us	0.000us	2.205s	10.42%	2.205s	73.510ms	0 b	0 b	0 b	0 b	30
Memcpy PtoP (Device -> Device)	0.00%	0.000us	0.00%	0.000us	0.000us	1.877s	8.87%	1.877s	6.953ms	0 b	0 b	0 b	0 b	270
void cutlass::Kernel<cutlass_80_tensorop_s1 688gemm_2...	0.00%	0.000us	0.00%	0.000us	0.000us	1.689s	7.98%	1.689s	112.608 ms	0 b	0 b	0 b	0 b	15
void cutlass::Kernel<cutlass_80_tensorop_s1 688gemm_1...	0.00%	0.000us	0.00%	0.000us	0.000us	1.677s	7.93%	1.677s	83.858ms	0 b	0 b	0 b	0 b	20
Self CPU time total: 27.501s														
Self CUDA time total: 21.160s														

Fig. 7 (Extended Data Fig. 13 in the revised manuscript). The CPU and GPU usage of training STACI for 6 epochs based on the 12 10x Visium samples of mouse brain coronal sections.

Two different usage samples at different epoch numbers are provided. The table lists the top ten functions by the total CUDA time ("self_cuda_time_total").

7. It is unclear how good the performance will be if using other spatial modalities like MERSCOPE and 10x genomics visium. Several available datasets are available, and the authors must show that the pipeline works with minimal changes from the user. A tutorial of such varied datasets should also be included in the github page.

We thank the reviewer for this comment. As discussed in response to point 2 and 6 above, we applied our method to a 10x Visium dataset with 12 samples and minimal changes were made to training the model. The Jupyter notebooks for running with the STARmap PLUS data and for running with the 10x Visium data are included in the Github repository.

8. How does STACI pipeline differ from the previously published work from the same group - cross-model autoencoders (Yang et al., Nat Comm, 2021). Can STACI be used to also integrate single-cell RNA and ATAC-seq data with spatial transcriptomics data?

While the integration of two modalities by matching the latent space of autoencoders in the STACI pipeline was inspired by the group's previous work, we would like to highlight two key methodological contributions of STACI over our previous work:

- a) The previous work provides a framework for integrating unpaired single-cell imaging and transcriptomic data. It cannot make use of spatial coordinates. STACI incorporates spatial information into the joint latent space using a graph autoencoder model. We demonstrated the importance of this new component in identifying biologically meaningful and disease-relevant tissue regions in Results Section 1.
- b) The previous work used a standard under-parameterized autoencoder without considering the problem of batch correction. A critical feature of STACI is the use of over-parameterization as a built-in method for batch effect correction in both gene expression and tissue morphology. To the best of our knowledge, over-parameterized autoencoders have never been used before in the biological context and our observation that they could be used for automatic batch removal is novel.

With the current setup described in the paper, paired datasets, which are standard in spatial transcriptomics, are required. But as also described in response to the reviewer's first comment, when additional modalities are available (such as scRNA-seq or scATAC-seq), our framework can easily integrate these by training one additional autoencoder per new modality and matching the latent space of the new autoencoder with the previously trained joint latent representation. The integration of unpaired single-cell datasets can be achieved by changing the l_2 loss used for matching the latent space of the two autoencoders as described in our current work to KL divergence described in Yang et al., Nat Comm, 2021², highlighting another difference between the two papers.

9. Can STACI be used to deconvolute cell-type composition in spot-based spatial transcriptomics datasets like 10x genomics visium platform. In theory, it should, but the authors need to explore it further.

We thank the reviewer for this interesting question. It is indeed possible to adapt STACI for deconvolution, since tissue images usually have subcellular resolution. To deconvolve spot-level measurements, the CNN autoencoder can be trained and fixed first, with the input image being individual cells. During the training of the graph autoencoder using the spot-level measurement, the latent representation of each spot can be matched to the latent representations of all the cells inside that spot so that the inferred individual cell expression adds up to the spot-level expression. This is part of ongoing work. A recent study explored a similar idea to deconvolute pixel-level gene expression with a different architecture⁸. Although this deconvolution task is out of the scope of the current project, we provide preliminary results from a different ongoing project where we deconvolute the 10x Visium measurements given H&E images in the context of intestinal fibrosis (see Fig. 8 below). Figure 8b highlights the group of genes including macrophage markers (FTH1 and CD68) and genes annotated in lipid metabolism (LIPA and APOC1). Adjusting the range of the input images, it is

also possible to differentiate between genes inside and outside of giant cells (Fig. 8c) which is a fibrosis-induced multi-nucleated cell without well-characterized gene expression markers.

Fig. 8 Deconvolution of 10x visium.

a) H&E of the intestinal fibrosis sample. b) The average activity of a group of genes with correlated gene expression patterns at each pixel. Higher intensity (brighter color) means higher expression. c) Cropping a small patch around a giant cell in a) results in better resolution of the estimated gene expression per pixel. Left most figure is the cropped H&E image used as the input. The other figures each represent the expression of one group of genes with correlated gene expression patterns. The first and second gene groups are able to identify the expression difference inside and outside of the giant cell.

Reviewer #3 Comments and Responses:

This work introduced a graph neural network model to combine gene expression, cell position and image modalities into a joint representation using the STARmap datasets. Authors provide a number of results but I am not sure they are evaluated in a proper way.

[Biological task]

There are a number of publications doing similar tasks but authors did not mention nor compare in the manuscript.

We thank the reviewer for pointing out the related works. We carefully updated the main text to include a paragraph in the introduction describing these works as well as how they differ from our proposed method (see below).

We would like to emphasize that a key conceptual and technical contribution of our method is that we only need to train one model for integrating all modalities with built-in batch correction to learn a joint latent representation of all modalities, which can be used for identifying biologically meaningful regions and enables the translation between different data modalities. This has several advantages compared to existing approaches:

- a) STACI learns a joint latent representation of gene expression, spatial location of cells, and chromatin images. The spatial location information is given by the physical neighborhood of cells, taking into account the expression of all measured genes in all cells in the neighborhood. Our setup of a separate decoder for each modality ensures that the latent space sufficiently captures the variations in all modalities. Learning a joint latent representation of gene expression and spatial location, instead of directly averaging gene expression in the original data space, allows for the integration of additional modalities and performing downstream tasks, such as batch correction and clustering, on all modalities. This would not be possible if the separation of cells into spatial domains is performed directly on the input data space.
- b) STACI corrects for batch effects in the joint latent space, which means that the correction is both in terms of gene expression and tissue morphology. To our knowledge, currently there is no method for correcting unwanted tissue morphology differences across samples. The correction of batch effects through over-parameterization of the autoencoder latent space does not make assumptions about the underlying distribution of the input data. Our approach only requires that the unwanted variations stemming from batch effects are smaller than the biological differences of interest, such as cell types and treatment effects.
- c) Our setup of using one autoencoder for each modality also enables the attribution of biological differences to each modality. We demonstrated this by identifying the association between heterochromatin and a cell's proximity to plaques.
- d) STACI can incorporate additional modalities by training a modality-specific autoencoder with a user-defined architecture, without retraining the existing autoencoders. This is valuable as datasets grow and more data modalities are becoming available.

In the following, we carefully discuss the differences between our approach and each related paper mentioned by the reviewer:

- For combining gene and spatial position:

StLearn: <https://www.biorxiv.org/content/10.1101/2020.05.31.125658v1>

stLearn normalizes gene expression by taking a weighted average of the gene expression of the spatial neighbors, where the weights are image similarity scores calculated as the cosine distances between the center spot and its neighbors. This is mainly a denoising approach that aims to reduce technical noise by smoothing the expression of a spot given neighboring spots that are expected to have similar gene expression according to image similarity. In contrast, STACI aims to identify patterns in cell neighborhoods, which can consist of diverse cell types/states. The latent representation of a cell takes into account all cells in the physical neighborhood regardless of

morphological similarity, which is more suitable for identifying meaningful tissue domains with consistent spatial patterns of cell states. Additionally, stLearn does not guarantee that the smoothing by image similarity retains all features of individual spots, since there is no mechanism for preventing oversmoothing. STACI ensures that the latent representations retain individual spot-level information while incorporating neighborhood information, by using a gene expression decoder to reconstruct the gene expression of each spot. At the same time, STACI reduces the impact of dropout by modeling gene expression as a zero-inflated negative binomial distribution; this has been shown in previous studies to reduce the impact of dropout⁹. Furthermore, STACI incorporates chromatin imaging data into the joint latent space, which as we demonstrated in Results Section 3-5 is useful for identifying morphological and molecular biomarkers of disease, and provides built-in batch correction of both gene expression and tissue morphology across samples.

In addition to the conceptual differences between the two approaches described above, we also added an empirical benchmark to compare the ability of STACI and stLEARN to identify tissue domains. We applied STACI to a 10x Visium dataset of mouse brain coronal sections available from the 10x website (<https://www.10xgenomics.com/resources/datasets/multiomic-integration-neuroscience-application-note-visium-for-ffpe-plus-immunofluorescence-alzheimers-disease-mouse-model-brain-coronal-sections-from-one-hemisphere-over-a-time-course-1-standard>); 10x Visium data of mouse brain coronal sections was also analyzed in the stLearn clustering tutorial (https://stlearn.readthedocs.io/en/latest/tutorials/stSME_clustering.html). As shown in Fig. 9a below, the stLearn clusters combine spots from different known anatomical regions of the brain: for example, cluster 18 contains cells from the hippocampus region, isocortex, olfactory areas, and cortical subplate; Cluster 4 contains both hypothalamus and cerebral nuclei; Cluster 15 is also scattered across thalamus, hypothalamus, and isocortex. Our results of applying the STACI model to the 10x Visium dataset of mouse brain coronal sections are shown in Fig. 4c (which is Extended Data Fig. 10c in the revised manuscript). This figure was also presented in the response to reviewer 1; we replicate it below for ease of reading. As seen in Fig. 4c, using STACI results in consistent clusters across all samples and the identified clusters correspond to known anatomical brain regions.

Fig. 9 Analysis of 10X Visium mouse brain coronal sections using stLearn and HMRf.

a) The clustering given by the stLearn method on 10x Visium data of mouse brain coronal sections (https://stlearn.readthedocs.io/en/latest/tutorials/stSME_clustering.html). b) A reference slice from the Allen Mouse Brain Atlas and Allen Reference Atlas - Mouse Brain. c) The clustering given by the HMRf method on 10x Visium data of mouse brain coronal sections (https://rubd.github.io/Giotto_site/articles/mouse_visium_brain_201226.html).

Fig. 4c (Extended Data Fig. 10 in revised manuscript). Validation of STACI on 10x Visium dataset of 12 mouse brain coronal sections.

Leiden clustering of the full STACI model with a latent dimension of 30000. The clustering resolution is 0.4. This clustering result shows consistent tissue segmentation across all samples and is consistent with known anatomical regions of mouse brains. In contrast, other existing methods for the analysis of spatial transcriptomics data result in having spots from different anatomical regions in the same clusters. For example, the tutorial of stLearn³ also shows an application to 10x Visium data of mouse brain coronal sections (https://stlearn.readthedocs.io/en/latest/tutorials/stSME_clustering.html), but one cluster contains cells from the hippocampus region, isocortex, olfactory areas, and cortical subplate (cluster 18) and another cluster contains spots from both hypothalamus and cerebral nuclei (cluster 4). The Giotto package's implementation of HMRF⁴ also demonstrates the results on 10x Visium data of mouse brain coronal sections (https://rubd.github.io/Giotto_site/articles/mouse_visium_brain_201226.html). While the accuracy and resolution of the spatial domains identified by HMRF and STACI are mostly comparable, some HMRF domains contain spots from different known anatomical regions of the brain in the same domains; for example, domain 16 contains cells from both inside and outside of the hypothalamus. Similarly, MUSE⁵, another existing method for the analysis of spatial transcriptomics data, shows mixing of cells from different brain regions into the same cluster (see Figure 6e⁵); for example, clusters 1, 4, and 15 show mixing of cells from the isocortex (CTX), hypothalamus (HY), and olfactory area (OLF), and cluster 0 contains a mixture of cells from the hippocampus and various other regions.

HMRF: <https://www.nature.com/articles/nbt.4260>

This method starts with selecting a set of spatially coherent genes, which are defined as genes that spatially separate cells with low expression and high expression of that gene. Cell-type specific genes are removed from the set of spatially coherent genes. For the resulting pre-selected set of genes and a pre-selected number of domains, an E-M algorithm is run to assign each cell to one of the domains based on its gene expression and the domain assignment of neighboring cells. Since HMRF does not use a latent representation of gene expression and cellular neighborhoods, it cannot be used to perform downstream tasks such as integrating images into the analysis of spatial transcriptomics data to predict gene expression from images and obtain morphological disease biomarkers. In addition, HMRF does not provide batch effect correction, especially for unwanted variations in tissue morphology.

To separate cells into a different number of domains or separate cells in the same domain to multiple subdomains using HMRF, the E-M algorithm needs to be rerun, whereas STACI learns the joint latent representation independently of the downstream task, such as clustering, which means that our model does not need any retraining to obtain more clusters or subclusters. In addition, STACI does not require the selection of spatially coherent genes. The graph autoencoder can automatically learn to assign weights to each gene that both preserve the neighborhood graph structure and the gene expression of each cell, which is guaranteed by using a decoder for gene expression and a decoder for the cell adjacency. Identifying spatial patterns using all genes through neural networks allows for the identification of patterns that involve multiple genes with concerted spatial variation, while each gene alone might not be spatially coherent. STACI also does not require the user to decide if and how many cell type specific genes to remove from the set of spatially coherent genes. The removal of cell type specific genes is used by HMRF to improve domain separation in the mouse visual cortex where all cell types tend to be mixed with each other. However, some tissue regions, such as DG and CA regions in mouse brains, have the same cell types localized in the same regions, in which case the cell type specific genes would largely overlap with spatially coherent genes. STACI does not require the user to make the choice of how much cell type information to retain by ensuring that both the gene expression and cell adjacency can be decoded through the joint latent space, regardless of the cellular organization in the particular tissue. If a user wants to subset genes by spatial coherence, STACI can also accommodate a truncated input gene expression matrix without changing the main architecture.

In addition to the conceptual differences described above, we added an empirical benchmark to compare the ability of STACI and HMRF to identify tissue domains. As already described above, we applied STACI to a 10x Visium dataset of mouse brain coronal sections available from the 10x website (<https://www.10xgenomics.com/resources/datasets/multiomic-integration-neuroscience-application-note-visium-for-ffpe-plus-immunofluorescence-alzheimers-disease-mouse-model-brain-coronal-sections-from-one-hemisphere-over-a-time-course-1-standard>). 10x Visium data of mouse brain coronal sections was also analyzed in the tutorial of the Giotto package that provides an implementation of HMRF (https://rubd.github.io/Giotto_site/articles/mouse_visium_brain_201226.html). As shown in Fig. 9c and 4c above, the accuracy and resolution of the spatial domains identified by HMRF and STACI are mostly comparable and correspond to the known anatomical regions of the brain (Fig. 9b). Some HMRF domains, however, contain spots from different known anatomical regions of the brain in the same domains, for example domain 16 contains cells from both inside and outside of the hypothalamus. Our results of applying the STACI model to the 10x Visium dataset of mouse brains is shown in Fig. 4c (Extended Data Fig. 10c in the manuscript). We obtained consistent clustering results across all samples and the identified clusters correspond to known anatomical brain regions.

- For combining image and gene:

MUSE: <https://www.nature.com/articles/s41587-022-01251-z>

MUSE integrates single-cell gene expression and images measured in spatial transcriptomics datasets, without utilizing the spatial locations of cells. MUSE first uses separate modality-specific encoders for gene expression and images. The encoder results are concatenated and passed through

additional neural network layers to obtain a joint latent representation. MUSE also has modality-specific decoders that decode gene expression and images from the joint latent space. Additionally, MUSE uses a triplet-loss function between the clustering of the output of each modality-specific encoder and the clustering of the joint latent representations. An important feature of STACI in contrast to MUSE is the integration of spatial coordinates as well as the built-in batch correction for both gene expression and tissue morphology.

As a result of the lack of spatial information, Figure 6e of the MUSE paper shows that multiple clusters contain a mixture of cells from different brain regions, even when the spatial transcriptomics dataset already provides averages of gene expression of the neighboring cells with the spot-level measurements. For example, clusters 1, 4, and 15 show mixing of cells from the isocortex (CTX), hypothalamus (HY), and olfactory area (OLF). Cluster 0 contains a mixture of cells from the hippocampus and various other regions. In contrast, as already described above, STACI applied to a spot-level spatial transcriptomics dataset of mouse brains available from the 10x website (<https://www.10xgenomics.com/resources/datasets/multiomic-integration-neuroscience-application-note-visium-for-ffpe-plus-immunofluorescence-alzheimers-disease-mouse-model-brain-coronal-sections-from-one-hemisphere-over-a-time-course-1-standard>) results in consistent clusters across all samples and the identified clusters correspond to known anatomical brain regions (Fig. 4c above).

In terms of integrating gene expression and images, there are major architectural differences between MUSE and STACI. In STACI, the structures of the modality-specific autoencoders are retained and are separate from each other. During training, the joint latent space of STACI is computed through matching the latent space of each autoencoder using l_2 loss. In contrast to MUSE, STACI does not necessitate concatenating the encoder outputs for the additional joint encoder layers before obtaining the joint latent space. These architectural differences of STACI have several advantages as compared to MUSE:

- a) The joint latent space of MUSE has to be obtained by using all input modalities. Thus, MUSE is unable to predict gene expression from images. Whereas we demonstrated in Results Section 3 that, at test time, the joint latent space of STACI can be computed from a single modality and that the other unmeasured modality can be predicted by applying the modality-specific decoder to the inferred joint latent space.
- b) When a new data modality becomes available, MUSE has to retrain the whole model, whereas STACI only needs to train one new autoencoder for the new modality and can match the latent space of the new autoencoder to the joint latent space of the other modalities without updating the previously trained autoencoders.
- c) For the same encoder and decoder structures, our setup enables separate training of each modality-specific autoencoder. This can significantly reduce the memory required during training, which is especially advantageous when the latent space is large.
- d) It is also unclear if MUSE can extract image features, such as heterochromatin changes, from the pretrained Inception model.

Stnet: <https://ai.stanford.edu/~bryanhe/publications/stnet.pdf>

The main purpose of ST-Net is to predict gene expression from H&E images. ST-Net uses CNN layers for the input images followed by a fully connected layer to predict gene expression. This architecture is similar to STACI in that STACI also uses a CNN encoder to encode images to the latent space and then decode gene expression from the latent space with a fully connected decoder. Thus, for the problem of predicting gene expression from H&E images, ST-Net and STACI are comparable, because the same CNN encoder and fully connected decoder structures can be used interchangeably in both methods. However, ST-Net is limited to two modalities with paired measurements on the same cells, but the framework of STACI can be used to incorporate more modalities and with unpaired measurements on different cells from the same population.^{1,2}

Other than predicting gene expression from images, ST-Net does not demonstrate the utility of its learned latent space in downstream analysis, except briefly mentioning that the tumor and normal spots can be separated in the latent space. Such separation could be obtained from a single modality of either gene expression or morphology. Most importantly, ST-Net does not incorporate spatial coordinates of cells and thus does not provide meaningful region segmentation given the latent space. Furthermore, in contrast to STACI, ST-Net does not provide a method for batch correction.

We updated the Introduction by adding the following paragraph to incorporate a careful comparison with these related works: "With the rise of spatial transcriptomic data, several computational approaches have been developed to integrate different data modalities in the tissue context. STACI is the first method, to our knowledge, that simultaneously integrates all the available modalities, namely gene expression, cellular neighborhoods, and chromatin imaging, and is capable of translating between different data modalities and identifying combined morphometric and molecular disease biomarkers in the tissue context. In particular, various methods have been developed that integrate or can be adapted to integrate single-cell gene expression with images through a joint latent space, but do not incorporate cell location into their analysis.³²⁻³⁴ In addition, other methods such as HMRF³⁵ incorporate spatial information into the analysis of gene expression to identifying spatial regions with consistent patterns of cell states but do not make use of imaging data. It is the joint latent representation of gene expression and cell location used by STACI that enables us to incorporate chromatin imaging data into the analysis and perform various downstream analysis, such as clustering cells into finer spatial regions, without retraining the model. An interesting method that integrates all three modalities available in spatial transcriptomic datasets is stLearn, a denoising approach that replaces the gene expression of a cell by the average expression of its neighboring cells, weighted by their image similarity.³⁰ In contrast, STACI aims to identify patterns in cell neighborhoods, which can consist of diverse cell types/states, by taking into account all cells in the physical neighborhood regardless of morphological similarity. Both stLearn and HMRF are unable to predict gene expression from images or identify morphological disease markers associated with the identified tissue regions. Finally, another key feature of STACI compared to current methods for multimodal integration is the built-in correction of batch effects that applies to all modalities in the joint latent space."

I believe there are also a number of other publications in this field.

- With that, some claims in the introduction are not accurate, such as "Technologies such as STARmap and 10x Genomics Visium measure chromatin staining paired with spatial transcriptomics, but current analysis methods do not make use of this information."

We thank the reviewer for this comment. Most of the previous methods either incorporate gene expression and images or incorporate gene expression and spatial information in the latent representation, but not all three modalities together. Even when all three modalities are used, such as in stLearn¹⁰, not all information from all modalities is utilized. For example, in stLearn, images are only used to estimate the similarity between cells. Whereas we demonstrated that informative disease-relevant features can be obtained by fully utilizing the available image data; e.g. we identified the heterochromatin ratio to be indicative of AD progression. We made the sentence more accurate as follows: "Technologies such as STARmap and 10x Genomics Visium measure chromatin staining paired with spatial transcriptomics, but current analysis methods do not make use of all three modalities together (chromatin staining, transcriptomics, and spatial coordinates) or only use images as a cell similarity metric for denoising the gene expression data³⁰."

- It would be valuable to benchmark the proposed model with existing works using datasets with "ground truth" to show the benefits.

We thank the reviewer for this comment. As described in our response to the reviewer's previous comment, to further benchmark our proposed model, we applied it to a 10x Visium dataset of

mouse brain coronal sections available from the 10x website (<https://www.10xgenomics.com/resources/datasets/multiomic-integration-neuroscience-application-note-visium-for-ffpe-plus-immunofluorescence-alzheimers-disease-mouse-model-brain-coronal-sections-from-one-hemisphere-over-a-time-course-1-standard>). This dataset has been widely used for benchmarking. In addition, benchmarking with mouse brain data also has the benefit that the known anatomical regions can be used as a ground truth (see Fig. 9b above). In addition to the results already described above in Fig. 4c, additional analysis of the resulting clusters can be found in Fig. 4-6 above (which correspond to Extended Data Fig. 10-12 in the revised manuscript). To describe this additional analysis, we added the following sentences to the Results Section 2: “We applied STACI to a 10x Visium dataset of 12 mouse brain samples consisting of AD and control mice at different time points⁶⁸. Compared to the clusters given by 10x based on gene expression alone (Extended Data Fig. 10a, 11), STACI achieves more consistent results across all samples, given the known anatomical regions of mouse brains (Extended Data Fig. 10c, 12).”

[Organization]

I have a major concern on this point.

- The manuscript is not in a good organization. Figures are arranged in an inconsistent way with the main text (especially for Figures 2-3 and ED figures). For the same result section, readers need to jump to different figures to see the related results. I think it might not be a good way to have one storyline in context but provide another different storyline in the figure organization.
- The description of methods is divided to several sections, which makes it hard to understand what conditions/frameworks were used to obtain the results. E.g. Fig 2b comes before introducing overparameterization and ZINB, but it is unclear this result was obtained with or without using these two elements.

We thank the reviewer for this comment. Below, we would like to explain the thinking behind our current organization of the figures. Since the other two reviewers did not suggest changes to the organization of the figures, we would prefer to keep the order as is. However, if the reviewer has a specific suggestion on how to rearrange a figure to make the story flow better, we would be very happy to reconsider. We feel that incorporating all the comments raised by all the reviewers as well as the additional benchmarking on the 10x Visium data and the additional validation on the new mouse samples has helped further strengthen the manuscript and made its structure more clear.

- Figure 1 is an overview of STACI and its applications, with the illustrations of the applications appearing in the same order as they appear in the main text. Figure 1a starts with an overview of the input data to STACI. Figure 1b then illustrates the overall architecture of STACI. Figure 1c and 1d are the two applications discussed later in the main text after discussing the architecture.
- Figure 2 demonstrates the application of STACI to the STARmap dataset and analyzes the different variants of the architecture, in terms of clustering and batch effect removal, two critical tasks for analyzing gene expression data. We begin by showing the results when using the overparameterized model with ZINB distribution. This is our proposed model and we feel it is important for the reader to upfront get an understanding of what STACI can achieve before going into a technical discussion, where we dissect each of the ingredients of STACI and show how the output would change by removing one of the ingredients. This discussion follows immediately after Figure 2b. We end the discussion in Figure 2 with the plaque distribution to put the clustering results back into the AD context to show that STACI can not only identify consistent tissue regions across samples, but the identified regions are also disease-relevant.
- Starting from Figure 3, the integration with chromatin images is demonstrated. Figure 3 focuses on the prediction of gene expression from chromatin images, which is also the first application we discussed in the main text after integrating the chromatin images. Figure 4 shows how we identified heterochromatin as the morphological disease marker. Figure 5 further examines heterochromatin changes and links heterochromatin with the earlier results

of the region-specific disease progression, which leads to the identification of disease marker genes. This figure demonstrates how we can improve the statistical power of marker gene identification by incorporating chromatin imaging.

- Each Extended Data Figure discusses a separate topic and we tried to divide the contents among the figures such that each figure can be understood by itself without referencing other figures. The order of the Extended Data Figures follows the same order as their respective content in the main text.

[Machine learning model]

It is really good to discuss the benefits from each architecture design.

- How to balance the reconstruction errors from each modalities. I think the reconstruction from image is quite different from expressions or positions.

We agree with the reviewer that it is important to carefully discuss each architecture design choice. We carefully dissected each architecture choice in Results Section 1 and 2, including removing one of the decoders or inputs, changing the statistical model of gene expression, changing the definition of cell adjacency, and removing over-parameterization.

We also thank the reviewer for the question regarding how to balance reconstruction errors from each modality. The graph autoencoder is trained and fixed first before training the CNN autoencoder and matching the CNN latent space to the graph autoencoder latent space. To analyze the balance between gene expression reconstruction and cell adjacency reconstruction, we plotted the training and validation losses for the full STACI model, the STACI model without gene expression reconstruction, and the STACI model without adjacency reconstruction (Extended Data Fig. 3a-c). As seen in these figures, adding the reconstruction loss of another modality (gene expression or cell adjacency) does not decrease the validation loss from training with the reconstruction loss of a single modality.

The design choice of training and fixing the graph autoencoder first before the CNN autoencoder instead of training the two autoencoders together was made because a previous study by Yang and Uhler¹ showed that training two autoencoders simultaneously significantly increases computational costs (as expected) but does generally not improve the performance of domain translation on a comparable task to predicting gene expression from images in our application (section 4.2, Figure 6 in Yang and Uhler 2019). In line with these prior observations, we observed that when training and fixing the graph autoencoder prior to the CNN autoencoder, the CNN autoencoder results in good reconstruction quality of the images (Extended Data Fig. 14). To further confirm this, we added Fig. 3 (which is Extended Data Fig. 15 in the manuscript) to show that the image reconstruction quality is comparable to training a CNN autoencoder alone without matching the latent space to the graph autoencoder. Thus, it is not necessary to update the graph autoencoder when training the CNN autoencoder to accommodate the CNN latent structure. While Fig. 3 was already shown above in response to a question by Reviewer 1, we provide it again below for ease of reading.

While this analysis shows that there are no disadvantages to fixing one autoencoder prior to the training of the other autoencoder, there are various benefits to doing so, including more stable training and less memory required for training, which is especially important in our over-parameterized model that has a larger memory requirement than the standard autoencoder. In addition, if there are additional modalities available to be incorporated into the joint latent space, the current setup where one autoencoder is fixed first does not require retraining of all the previously trained autoencoders.

Fig. 3 (Extended Data Fig. 15 in revised manuscript). Examples of input chromatin images to the CNN autoencoder and the corresponding reconstructed images. There is no latent space matching with the graph autoencoder. All samples were used in training the CNN autoencoder.

- Current comparisons are among variants from the same GNN framework. Finding an optimal framework under GNN cannot guarantee a comparable performance comparing with existing tools in the field.

We thank the reviewer for this comment. As described in response to the first comment by the reviewer, we added a paragraph in the introduction, where we review existing tools for the analysis of spatial transcriptomics data and carefully discuss how they differ from our proposed method. Please see our responses to your first question.

- Experiment conditions: due to the organization, it is unclear what settings were used for each experiment.

We provided a point-by-point reply to the specific comments below.

- Evaluations: (as described below), current metrics are not objective and cannot be viewed as ground truth.

We provided a point-by-point reply to the specific comments below.

- As authors said, it is unclear how overparameterization works. I think it is risky to make hypothesis that it can reduce batch bias and emphasize biological signals, as it has no information about the difference between batch and true signal.

We thank the reviewer for these important comments. Batch correction is an important problem in the field. We attempted to work on this by providing a method based on over-parameterization. While there could certainly be a deeper theoretical understanding of the method, we believe we have sufficient theoretical understanding to know when the method will and will not work. Interestingly, a principal component analysis of the dataset can be used to assess quantitatively whether over-parameterization would be effective. Namely, the method will work if the actual biological differences of interest but not the batch differences correspond to the top principal components of the data. Our group provided a theoretical proof that over-parameterization stretches the data along the top principal components for very simple neural network architectures.¹¹ We demonstrated empirically that this result applies to the autoencoder model used in STACI: namely, in Extended Data Fig. 6, we showed that over-parameterization of the graph autoencoder stretches the STARmap data along the top principal components. Although there is no explicit information of which difference is due to batch effect, one can check through PCA that the top principal components do not correspond to batch differences, in which case over-parameterization is appropriate for batch effect removal. Motivated by another question by the reviewer below, we now also show that over-parameterization can be used for batch effect removal in single-cell RNA-seq data when the top principle components do not correspond to batch differences (see our response below to the specific question of the reviewer on using over-parameterization for batch effect removal in single-cell RNA-seq data). In summary, we feel that over-parameterization is a novel and appealing approach for batch effect removal that can easily be integrated into any autoencoder framework and we understand it well enough to provide quantitative guidance on when to use the approach.

[Data analysis]

- For image feature learning, as I can tell from STARmap technology, you can actually extract exact single cell images using segmentation masks. Will this be more accurate than the squared segmentation?

For learning features in the latent space, the exact single-cell information has been incorporated into the latent representation by encoding and reconstructing the gene expression matrix which has single-cell resolution. Matching the CNN latent space to the graph autoencoder latent space automatically ensures that single-cell information is extracted and additional image segmentation is not necessary. In fact, we chose the squared segmentation of the chromatin images to have a field of view matching the size of the neighborhood used in the graph convolution. This is described in Results Section 4: "To do this, we train a fully connected neural network on the joint latent space to predict the size of plaques in an image patch, which encompasses a similarly sized neighborhood as the

20-nearest-neighbors used to form the adjacency matrix for the spatial transcriptomics autoencoder, centered at each nucleus”.

[Experiments]

Batch effects:

- In Fig 2g, h, I don't think there is a strong link between these 3 clusters and plaques, as you also identified three similar clusters from control mice. I suspect this is a consequence from different subregions. Another way to check is to perform differential expression analysis among these regions and see what comes out.

We thank the reviewer for this comment. It is correct that the clusters are identified without using the plaque images and they are subregions that exist in all mice. The separation of these clusters in the graph autoencoder latent space suggests that the difference between the clusters is in the distribution of cell states in the three regions. Our results further suggest that some of these regions could be more susceptible to plaque accumulation than others. An analogy is that plaques tend to accumulate more in the cortex than in the CA region. Similarly as for the 3 regions that we identified, the cortex and the CA regions can be distinguished without the use of plaque and just using anatomical and gene expression differences in healthy mice.

As proposed by the reviewer, we performed differential expression analysis among the three regions for the three cell types we analyzed in Fig. 5 in the main text, excitatory neurons, microglia, and oligodendrocytes. The results are shown in Fig. 10 below and were added as Extended Data Fig. 28 into the revised manuscript. The differentially expressed genes show consistent spatial distribution as expected by our analysis in Fig. 5b of the main text and by the genes known to associate with AD. For example, from cluster 1 to cluster 2 and 3, *Ctsb* in microglia increases in average expression and in the fraction of cells expressing the two genes. This is consistent with the higher expression in the high-gradient high-heterochromatin microglia in the 13-month AD sample compared to the low-gradient low-heterochromatin microglia in the 13-month control sample (Figure 5b in the main text). *Ctsb* is also known to be up-regulated in the disease-associated subtype of microglia¹². We added the following sentence to Results Section 5 summarizing these findings: “These marker genes of microglia also show a consistent spatial distribution across the three cortex regions we identified, e.g. *CTSB* is increasingly up-regulated from cluster 1 to cluster 2 and 3 (Extended Data Fig. 28).”

Importantly, we were able to obtain 4 additional mice samples that were measured with the same STARmap PLUS technology to validate the 3 clusters that we identified in the brain cortex. These additional samples confirmed that the three clusters are consistent across mice samples: The three cortex regions were identified in all four samples; see Fig. 11a below (which is Extended Data Fig. 2a in the revised manuscript) and we also found the same trends in terms of plaque distribution in the new samples with cluster 3 having more and larger plaques than cluster 1 (see Fig. 11b below). These results further validate the link between the three cortex clusters and plaque distribution. The following description of the new samples were added to the Results: “We applied the trained STACI model and the same analysis to four new mouse samples held out for validation. All the resulting tissue segmentations are consistent in the new samples, including the separation into 3 clusters in the cortex regions with cluster 1 consistently having smaller plaques than the other two cortex clusters (Extended Data Fig. 2).”

Fig. 10 (Extended Data Fig. 28 in the revised manuscript). Differential expression analysis of the three cortex regions identified by STAC1.

Statistical significance was defined as p-value < 0.05 after correction by Benjamini-Hochberg procedure and fold change of at least 10% in either direction. At most 15 genes with smallest p-values are shown.

Fig. 11 (Extended Data Fig. 2 in the revised manuscript). Validation of STACI on four held-out mouse samples shows consistent tissue segmentation and plaque size distribution in the cortex.

a) The full STACI model with 6000 latent dimensions was applied to the four samples held-out in the original analysis. All eight samples were clustered using Leiden with a resolution of 0.82.

b) Histograms of plaque size, measured in number of pixels, are plotted for the cortex regions of the two AD samples. Frequency is normalized by the area of each cortex region. Consistent with the findings in the original four mice samples, larger plaque sizes are observed in cluster 3 as compared to cluster 1 in the 8-month held-out AD mouse (p-values: $1.5e-30$ for cluster 3 vs cluster 1) and are observed in both cluster 2 and cluster 3 as compared to cluster 1 in the held-out 13-month AD mouse (p-values: $5.5e-9$ for cluster 3 vs cluster 1 and $6.4e-15$ for cluster 2 vs cluster 1).

- Following the above point, there is no clear evidence that these 3 regions are meaningful in biology. Given the small number of plaques, the statistical test is not reliable. I think it is less reasonable to compare results based on these 3 clusters.

We thank the reviewer for this comment. As described in Section 1 of Results, the three cortex clusters correspond to “the outer layers of the primary somatosensory area (cluster 1), inner layers of both the somatomotor area and the primary somatosensory area (cluster 2), and both the retrosplenial area and the outer layers of the somatomotor area (cluster 3).” Furthermore, as described also in answering the previous comment, we were able to provide additional evidence that the 3 regions are meaningful biologically by confirming that the three clusters can be consistently reproduced also in another set of 4 mice. In particular, we were able to reproduce the three cortex regions in all four additional samples (see Fig. 11a above) and we found the same plaque distribution also in the new mice samples with cluster 3 having more and larger plaques than cluster 1 (see Fig. 11b above).

In addition to the difference in plaque, we also showed that the three regions differ from each other in heterochromatin features as well as regression gradient in all cell types. In particular, there is a consistent shift to higher heterochromatin and higher gradient from cluster 1 to the other clusters. Finally, the plaque induced change as measured by the differential expression analysis also indicates meaningful marker genes (Fig. 5b in the main text).

- ED Fig 3: spatial coherence of clusters cannot be used as a ground truth without additional evidence -- as we know some cell types are distributed randomly in brain.

Brains can be consistently defined anatomically and some anatomical regions have distinct cell type distributions that can be identified without additional computational algorithms. One such example is the highly organized hippocampus region, which is known to divide into CA1, CA2, CA3, DG regions, and their sub-domains both anatomically and according to gene expression.¹³⁻¹⁷ Thus, it is at least expected that these regions that are distinct in cell types and anatomy should be separated in clustering by using either gene expression alone or gene expression with cellular adjacency. We did observe that with gene expression alone as the input, the hippocampus regions are separated according to the Allen Brain Atlas (Extended Data Fig. 4e, Fig. 2f in the main text), which verified that our model correctly captures variations of gene expression across cell types. If we add cell adjacency as the input but do not use an adjacency decoder, the CA3 cluster also contains cells in the CA1 region and the cells are not adjacent in space (Extended Data Fig. 4c). Similarly, if both gene expression and cell adjacency are used as the input without using any decoder, the CA regions are not separated according to the known anatomical and gene expression differences, but are separated into a cluster containing part of the CA1 region and another cluster containing the rest of the CA1 and the CA3 region (Extended Data Fig. 4d). If using the full STACI model but removing the zero-inflation dropout rate parameter in the gene expression decoder, CA3 is clustered with corpus callosum and some other regions of the hippocampus (Extended Data Fig. 4f). Only when the full STACI model is used, the ground truth separation in the hippocampus region can be recovered (Fig. 2b in the main text).

While in the isocortex region one cannot identify a clear separation of cell types into different regions by eye¹⁷, our goal was to develop a computational method that can find patterns in the distribution of cell types in space and validate the patterns with orthogonal measurements such as heterochromatin ratio and plaque distribution. We demonstrated in Fig. 11 above (which is Extended Data Fig. 2 in the revised manuscript) that the three cortex clusters that we had identified and the differences in plaque distribution in these three clusters are reproducible also in the four newly generated samples that have not been used in our previous analysis.

Finally, for further validation of our method on a different dataset obtained with a different technology, we also applied STACI to 10x Visium samples of mouse brains, demonstrating that the resulting clusters are consistent with the major anatomical regions in the Allen Brain Atlas (see Fig. 4c and Fig. 9b above).

- In ED Fig 4, as 4 tissue slices are from different biological conditions, we do not 100% sure they should be overlapped completely -- additional evidence is needed.

We thank the reviewer for this comment and agree that subtle differences should be expected between the different biological conditions. However, both the 8-month and 13-month mice are mature and we should expect the major anatomical regions of the brain to be the same across the two mice and disease conditions, such as the CA and DG regions. In accordance, the brain regions identified by STACI are reproducible in the four newly generated samples without sample-to-sample differences, indicating that there are consistent patterns of cell type/state distribution across samples at the level of resolution of anatomical regions considered in our paper. But as shown in our paper, a separate analysis of each of the identified clusters shows how gene expression as well as chromatin features change within a cluster from 8 to 13 months and from diseased to non-diseased (Fig. 5 in the main text).

- To test the model on batch bias removal, authors can use multiple scRNAseq data sets with batch labels and check how the overparameterization works to remove the bias.

We thank the reviewer for this suggestion. As per the reviewer's suggestion, we tested our method on an scRNA-seq dataset which has previously been used for benchmarking batch correction methods^{18,19}. A principal component analysis shows that the main principal components are driven by cell type differences and not by batch differences (see Fig. 12a below), which as explained in response to another question by Reviewer 3 above, suggests over-parameterization as an appropriate method for batch effect removal in this dataset. As suggested by Reviewer 3 below, we used entropy to quantitatively assess the mixing of the batches and the preservation of the cell types in the over-parameterized latent space as compared to the original data space. As shown in Figure 12b-e below, the entropy of mixing increased with respect to the batch labels (better mixing) and decreased with respect to the cell type labels (better cell type separation) using over-parameterization. As expected based on our theoretical understanding of the effect of over-parameterization (described above in response to another question by Reviewer 3), this correction of batch effects is a result of stretching the data along the top principal components in the over-parameterized latent space, which are dominated by cell type differences, and relatively reducing the variance in the lower principal components, which contain the batch differences (see Fig. 12f-g below).

Fig. 12. Over-parameterization removes batch effects in scRNA-seq data

- a) PCA of the scRNA-seq dataset of bone marrow^{18,19}.
- b) UMAP and entropy of mixing of the original gene expression data given the batch labels.
- c) UMAP and entropy of mixing of the original gene expression data given the cell type labels.
- d) UMAP and entropy of mixing of the latent space of STACI given the batch labels. No adjacency input is used for this scRNA-seq dataset. The latent dimension is 30000.
- e) UMAP and entropy of mixing of the latent space of STACI given the cell type labels. No adjacency input is used for this scRNA-seq data. The latent dimension is 30000.
- f) The variance explained by each principal component of the original gene expression data.
- g) The variance explained by each principal component of the STACI latent representation using a latent dimension of 30000.

- There are a number of approaches handling the batch bias problem, can you compare with them?

To our knowledge, current methods for the batch bias problem do not incorporate batch correction for tissue morphology and only account for gene expression differences across batches. As we demonstrated in Extended Data Fig. 6a and 6e, the sample-to-sample variation is less significant if cell location is not incorporated into the latent representation. Without cell location in the latent representation, clustering is no longer dominated by differences between samples as shown by the heatmaps. Thus, sample-to-sample difference in tissue morphology contributes significantly to batch bias and requires new methods to deal with it. We attempted to provide a first method that could perform batch correction in the context of tissue morphology.

- Visualization is not a good measurement of batch bias. You might want to use quantitative metric to measure them, such as entropy of mixing

We thank the reviewer for this comment and suggestion. In response to this comment, we computed both the entropy of mixing and the silhouette score, which is a metric used in previous studies¹⁹⁻²¹, to quantitatively assess the amount of batch effect removal. We did this for each cluster given the sample labels. The absolute values of the resulting silhouette scores as well as the entropy are shown in Fig. 13 below, which we added as Extended Data Fig. 7a in the revised manuscript. For the silhouette scores, a value closer to 1 means less sample-to-sample variation. With over-parameterization (latent space dimension = 6000), the silhouette scores are much higher than the same model without over-parameterization (latent space dimension = 1024), using the same Leiden clustering resolution at 0.1. Even when increasing the resolution for the over-parameterized model to obtain more clusters, this still resulted in higher silhouette scores than the under-parameterized model. The entropy for each cluster shows a similar improvement when going from the under-parameterized model to the over-parameterized model: while in the under-parameterized model, the entropy is 0.79, it increases to 1.12 in the over-parameterized model, indicating better mixing of batches.

We added the following sentences reflecting this analysis to Results Section 2: "Assessing batch effects using the average silhouette width, a measure used in previous studies,⁴⁷⁻⁴⁹ also indicates that over-parameterization leads to a significant reduction of batch effects (Extended Data Fig. 7a, Methods). This improvement in batch separation is further confirmed by using entropy of mixing, which improved from 0.79 in the under-parameterized model to 1.12 in the over-parameterized model and shows consistent improvements in each cluster (Extended Data Fig. 7a, Methods)."

We also added a description of how the two metrics were computed to the Methods section: "**Metrics of batch effect correction:** The silhouette score is computed using the `silhouette_batch` method in the `scib` package.⁴⁹ The entropy of mixing follows the implementation in a previous study.⁴⁸ Given 100 randomly selected cells from all four samples, the entropy of each cell is computed given the sample label of the 50 nearest cells in the latent space. The average entropy over the 100 randomly selected cells was averaged over 100 bootstrap samples. The entropy of mixing is calculated for all cells and also for cells in each cluster separately."

Fig. 13 (Extended Data Fig. 7a in the revised manuscript). Over-parameterization removes batch effects.

a) Quantification of mixing of cells from different samples in the under-parameterized latent space and the over-parameterized latent space for each cluster. Each dot represents the score of one cluster. The numbers (0.1 or 0.2) indicate the clustering resolutions while all other clustering parameters are kept the same. Left: average silhouette width. Right: entropy of mixing. For both metrics, higher score means better mixing and less batch effects.

Gene prediction:

- Fig 3a, predicting the expression of 8 month control sample using models trained on other samples
- will biological differences affect the result?

Since gene expression and genome packing are tightly linked, we can expect the biological differences in gene expression to be reflected in the biological differences in chromatin images. By using the 8-month control, 13-month control as well as 13-month AD mouse as the training set, we expect the training set to cover the relevant variations in gene expression and morphology present in the 8-month AD mouse that is used as the test example. Our prediction results showed that we can identify cell types from the predicted gene expression without prior knowledge on the cell type annotations. The clustering of the predicted gene expression also recovered known cell type markers as the cluster markers through differential expression of the predicted gene expression. Thus, based on our prediction results, we can conclude that the cell type identification and major cell type markers are not affected by the biological differences, indicating that STACI can be used to generate hypotheses based solely on the predicted gene expression patterns.

Plaque size prediction:

- I think this section did not illustrate the benefits from the model. If most informative features are from image, will image alone analysis are more meaningful instead of a combined analysis?

We thank the reviewer for this question. The combined analysis, with gene expression and cell adjacency, was used to define the three cortex regions. Using the images alone but leaving out the transcriptomic data would have resulted in a less informative latent space embedding, which for example would not have the resolution to identify the three cortex regions. A less informative latent space embedding would also have negatively impacted the regression analysis to identify subtle imaging biomarkers of disease. In addition, a combined latent space is necessary to link the imaging features to molecular features of disease as we did in Fig. 5 in the main text. As we showed, the joint embedding allows for the identification of gene expression biomarkers with more statistical power than would have been possible if only one of the modalities were used.

[Tool]

- Link provided is invalid.

In the initial submission we had provided all code related to this manuscript in a zip file. We thank the reviewer for pointing out the GitHub link issue. The repository is now public.

References

1. Yang, K. D. & Uhler, C. *Multi-Domain Translation by Learning Uncoupled Autoencoders*. <http://arxiv.org/abs/1902.03515> (2019) doi:10.48550/arXiv.1902.03515.
2. Yang, K. D. *et al.* Multi-domain translation between single-cell imaging and sequencing data using autoencoders. *Nat. Commun.* **12**, 31 (2021).
3. Haase, R. *et al.* CLIJ: GPU-accelerated image processing for everyone. *Nat. Methods* **17**, 5–6 (2020).
4. Eng, C.-H. L. *et al.* Transcriptome-scale super-resolved imaging in tissues by RNA seqFISH+. *Nature* **568**, 235–239 (2019).
5. Wang, X. *et al.* Three-dimensional intact-tissue sequencing of single-cell transcriptional states. *Science* **361**, (2018).
6. Xia, C., Fan, J., Emanuel, G., Hao, J. & Zhuang, X. Spatial transcriptome profiling by MERFISH reveals subcellular RNA compartmentalization and cell cycle-dependent gene expression. *Proc. Natl. Acad. Sci.* **116**, 19490–19499 (2019).
7. Fawkner-Corbett, D. *et al.* Spatiotemporal analysis of human intestinal development at single-cell resolution. *Cell* **184**, 810–826.e23 (2021).
8. Bergenstråhle, L. *et al.* Super-resolved spatial transcriptomics by deep data fusion. *Nat. Biotechnol.* **40**, 476–479 (2022).
9. Eraslan, G., Simon, L. M., Mircea, M., Mueller, N. S. & Theis, F. J. Single-cell RNA-seq denoising using a deep count autoencoder. *Nat. Commun.* **10**, 390 (2019).
10. Pham, D. *et al.* stLearn: integrating spatial location, tissue morphology and gene expression to find cell types, cell-cell interactions and spatial trajectories within undissociated tissues. 2020.05.31.125658 Preprint at <https://doi.org/10.1101/2020.05.31.125658> (2020).
11. Jain, S., Radhakrishnan, A. & Uhler, C. A Mechanism for Producing Aligned Latent Spaces with Autoencoders. *ArXiv210615456 Cs* (2021).
12. Keren-Shaul, H. *et al.* A Unique Microglia Type Associated with Restricting Development of Alzheimer's Disease. *Cell* **169**, 1276–1290.e17 (2017).
13. Thompson, C. L. *et al.* Genomic Anatomy of the Hippocampus. *Neuron* **60**, 1010–1021 (2008).
14. Datson, N. A. *et al.* Expression profiling in laser-microdissected hippocampal subregions in rat

- brain reveals large subregion-specific differences in expression. *Eur. J. Neurosci.* **20**, 2541–2554 (2004).
15. Lein, E. S., Zhao, X. & Gage, F. H. Defining a Molecular Atlas of the Hippocampus Using DNA Microarrays and High-Throughput In Situ Hybridization. *J. Neurosci.* **24**, 3879–3889 (2004).
 16. Lein, E. S., Callaway, E. M., Albright, T. D. & Gage, F. H. Redefining the boundaries of the hippocampal CA2 subfield in the mouse using gene expression and 3-dimensional reconstruction. *J. Comp. Neurol.* **485**, 1–10 (2005).
 17. Yao, Z. *et al.* A taxonomy of transcriptomic cell types across the isocortex and hippocampal formation. *Cell* **184**, 3222–3241.e26 (2021).
 18. Oetjen, K. A. *et al.* Human bone marrow assessment by single-cell RNA sequencing, mass cytometry, and flow cytometry. *JCI Insight* **3**, 124928 (2018).
 19. Luecken, M. D. *et al.* Benchmarking atlas-level data integration in single-cell genomics. *Nat. Methods* **19**, 41–50 (2022).
 20. Rousseeuw, P. J. Silhouettes: A graphical aid to the interpretation and validation of cluster analysis. *J. Comput. Appl. Math.* **20**, 53–65 (1987).
 21. Haghverdi, L., Lun, A. T. L., Morgan, M. D. & Marioni, J. C. Batch effects in single-cell RNA sequencing data are corrected by matching mutual nearest neighbours. *Nat. Biotechnol.* **36**, 421–427 (2018).

Reviewers' Comments:

Reviewer #1:

Remarks to the Author:

The authors have addressed all of my concerns in the revision satisfactorily.

Reviewer #2:

Remarks to the Author:

The authors have satisfactorily addressed all the comments and I recommend the article for immediate publication

Reviewer #3:

Remarks to the Author:

I thank authors' reply to my comment. I have following questions:

[Method comparison] we can always claim the difference from existing approaches. But a major point in computational paper is to justify how your new design could help biology comparing existing approach, otherwise this can be a "modeling for modeling" case. For example, authors mentioned one important property is the proposed model is built-in batch correction. Will this guarantee a better performance comparing with standard batch removal + feature learning?

During the paper review period, another paper is out for graph based integration of >2 modalities <https://www.nature.com/articles/s41587-022-01284-4>

[How to balance the reconstruction errors from each modalities]

I was thinking about what if you transform the gene data or normalize/scale them in a different way, will this result a dramatical change in joint training?

[Code]

I check the github and saw experiments with "convolution exploded"

(https://github.com/uhrerlab/STACI/blob/master/train_jointGAEcnnRegrs_starmap_multisamplesMixed.ipynb) "name errors"

(https://github.com/uhrerlab/STACI/blob/master/train_jointGAEcnn_starmap_multisamples.ipynb)

This is not something I expect from github repository. I was thinking can you provide a demonstration for readers showing how to use your tool?

Reviewer #1 (Remarks to the Author):

The authors have addressed all of my concerns in the revision satisfactorily.

We thank the reviewer for the positive feedback.

Reviewer #2 (Remarks to the Author):

The authors have satisfactorily addressed all the comments and I recommend the article for immediate publication

We thank the reviewer for the positive feedback.

Reviewer #3 (Remarks to the Author):

I thank authors' reply to my comment. I have following questions:

[Method comparison] we can always claim the difference from existing approaches. But a major point in computational paper is to justify how your new design could help biology comparing existing approach, otherwise this can be a “modeling for modeling” case. For example, authors mentioned one important property is the proposed model is built-in batch correction. Will this guarantee a better performance comparing with standard batch removal + feature learning?

We thank the reviewer for this comment. We showed in our manuscript that STACI enabled us to gain biological insights into the region-specific progression of Alzheimer's Disease, using STARmap data. We identified a novel separation of the mouse cortex into three regions that have different amounts of amyloid-beta plaque accumulation. In our previous revision, we also demonstrated that the three cortical regions and the corresponding difference in plaque accumulation can be consistently reproduced in four newly acquired validation samples (Extended Data Fig. 2). Analysis of the joint latent space of STACI allowed us to identify the heterochromatin ratio in cells as a highly predictive biomarker of nearby plaque accumulation and to identify gene expression changes corresponding to the change in heterochromatin ratio and the accumulation of plaque. In addition to the new biological insights enabled by STACI, the analysis we performed in the previous revision showed that STACI can also be applied to visium data and accurately segmented all samples in a 10x Visium dataset of mouse brain coronal sections¹ according to the known anatomical regions (Extended Data Figure 10c). In summary, we presented a new method that we believe will enable researchers to obtain novel biological hypotheses about the interplay between gene expression and chromatin condensation patterns during disease progression in complex microenvironments and identify novel disease biomarkers that capture all the available modalities.

In response to the reviewer's comment, we performed an additional comparison between our built-in batch removal method through over-parameterization and two popular batch removal methods, mutual nearest neighbors (MNN)² and ComBat^{3,4}. We used these two methods to pre-process a publicly available 10x visium dataset of mouse brain coronal sections¹, where the known anatomical regions can be used as a ground truth tissue segmentation. This dataset was suggested by the reviewers in the first revision and is the same dataset that we used in the first revision to demonstrate the applicability of STACI to large-scale visium datasets (Extended Data Fig. 10). We trained an under-parameterized version of the STACI model with a reduced latent dimension size separately on the data pre-processed with MNN as well as ComBat. While we provide more details about this analysis below, in summary we found that the full STACI model with over-parameterization has a better performance in terms of training stability as well as agreement of the resulting clusters with known anatomical segmentations than the use of standard batch removal methods such as MNN or ComBat applied with an under-parameterized model. This is summarized in Figure 1 below (which we added as Extended Data Fig. 13 to the manuscript) and summarized with the following text that we added to the manuscript: "STACI also achieves better results in terms of consistency across samples and consistency with the known anatomical regions as compared to applying the same architecture with an under-parameterized latent space to input data batch corrected by mutual nearest neighbors (MNN)⁴⁸ or ComBat^{53, 54} (Extended Data Fig. 13)."

In the following, we provide more details of the analysis. We trained an under-parameterized model with 32 latent dimensions (similar to the deep count autoencoder method⁵ for feature learning from scRNA-seq data) separately on the gene expression data pre-processed by the two batch correction methods (MNN and ComBat). For the MNN method, we applied library-size normalization, log transformation, and selected highly variable genes as the input to the method as described in the MNN paper². For the ComBat method, we followed the same procedure as Hu *et al.*⁶, i.e., the raw data was library-size normalized, log transformed, z-score normalized, and batch corrected by the ComBat method. As shown in Figure 1 below, the results using the MNN method together with an under-parameterized model results in clusters that do not correspond accurately to the known anatomical regions as well as clusters that are not consistent across samples. Using data pre-processed by ComBat as input to the under-parameterized model leads to unstable training: the gradient or loss grows to infinity in the first few epochs, even after reducing the learning rate from 0.01 to 0.0001. The training output of the ComBat method is provided in the notebook "train_gae_visium_10xADDFPE_extBatchCorrection.ipynb" in the github repository. The corresponding results using the STACI model with over-parameterization (and no separate pre-processing of the data for batch effect removal) are shown in Extended Data Fig. 10c showing that the resulting clusters are consistent with known anatomical regions as well as consistent across all samples. These results provide an additional demonstration that the over-parameterization in STACI results in more stable training as well as more biologically meaningful latent representations with less batch effects.

We would also like to reiterate the additional analysis that we had carried out in our previous revision, which also showed that STACI provides more accurate tissue segmentation than other methods including stLearn⁷, HMRF⁸, or MUSE⁹, which were suggested by the reviewer. In order to compare STACI to the best results of the three existing methods in terms of hyperparameter choices, we compared STACI to the results of the three methods provided in their tutorials or in their paper. All comparisons were made on spot-level measurements of mouse brain coronal sections (10x visium for HMRF and stLearn, and Spatial Transcriptomics for MUSE). Given the known anatomical regions (Figure 2a below), STACI's segmentation of the mouse brain (Extended Data Fig. 10c, reproduced in Figure 2b below) is more accurate than the results of stLearn (Figure 2c), HMRF (Figure 2d), or MUSE (Figure 6e of the MUSE paper, reproduced as Figure 2e below). The stLearn clusters combine spots from different known

anatomical regions of the brain: for example, cluster 5 contains cells from the hippocampus region, isocortex, olfactory areas, and cortical subplate (highlighted in red boxes in Figure 2c); Cluster 4 contains both hypothalamus and cerebral nuclei; Cluster 15 is also scattered across thalamus, hypothalamus, and isocortex (Figure 2c). Similarly, some HMRF domains contain spots from different known anatomical regions of the brain in the same domains; for example, domain 16 contains cells from both inside and outside of the hypothalamus (Figure 2d, black boxes). As a result of the lack of spatial information, clustering by MUSE shows that multiple clusters contain a mixture of cells from different brain regions, even when the spatial transcriptomics dataset already provides averages of gene expression of the neighboring cells with the spot-level measurements. For example, clusters 1, 4, and 15 show mixing of cells from the isocortex (CTX), hypothalamus (HY), and olfactory area (OLF). Cluster 0 contains a mixture of cells from the hippocampus and various other regions.

a. Underparameterized model with standard preprocessing and batch correction

b. Cells in cluster 0 (blue) of the underparameterized model with standard processing

Figure. 1 (Extended Data Fig. 13 in the revised manuscript). Application of under-parameterized STACI

model on 10x Visium dataset of 12 mouse brain coronal sections¹, where the gene expression data was pre-processed by mutual nearest neighbors (MNN)² for batch effect removal.

a) Leiden clustering (with a resolution of 0.8) of the resulting latent space when training an under-parameterized STACI model with a latent dimension size of 32 and 17,186 input genes. Before using the pre-processed data in the autoencoder model, we applied library-size normalization, log transformation, and selected highly variable genes as the input to the MNN batch correction method as described in the MNN paper².

b) Cells in cluster 0 of the clustering result described in a) are highlighted in blue, all other cells are colored in orange, showing that the tissue regions included in cluster 0 are not consistent across the different samples and that cluster 0 does not correspond to a well-defined anatomical region in any of the samples.

c) A reference slice from the Allen Mouse Brain Atlas and Allen Reference Atlas - Mouse Brain¹⁰ showing approximately the same anatomical region as in the 10x Visium dataset.

Figure. 2 Analysis of 10X Visium mouse brain coronal sections¹ using stLearn⁷, HMRF⁸ and MUSE⁹.

- a) A reference slice from the Allen Mouse Brain Atlas and Allen Reference Atlas - Mouse Brain¹⁰ showing approximately the same anatomical region as in the 10x Visium dataset. Black boxes highlight brain regions that contain cells in cluster 16 of the HMRF method, as in d).
- b) STACI clustering of the 10x Visium data of mouse brain coronal section¹.
- c) The clustering given by the stLearn method⁷ on 10x Visium data of mouse brain coronal sections (https://stlearn.readthedocs.io/en/latest/tutorials/stSME_clustering.html). Red boxes highlight brain regions that contain cells in cluster 5, which does not correspond to a well-defined anatomical region but contains cells in the hippocampus region, isocortex, olfactory areas, and cortical subplate
- d) The clustering given by the HMRF method⁸ on 10x Visium data of mouse brain coronal sections (https://rubd.github.io/Giotto_site/articles/mouse_visium_brain_201226.html). Black boxes highlight cells in cluster 16, indicating that cluster 16 has cells both inside and outside of the hypothalamus. The same regions are also highlighted in black in the reference slide in a)
- e) The clustering given by the MUSE method⁹ on Spatial Transcriptomics data of mouse brain. This is Figure 6e of the MUSE paper. Clusters 1, 4, and 15 show mixing of cells from the isocortex (CTX), hypothalamus (HY), and olfactory area (OLF). Cluster 0 contains a mixture of cells from the hippocampus and various other regions. The clustering result is not consistent with the known anatomical regions.

During the paper review period, another paper is out for graph based integration of >2 modalities
<https://www.nature.com/articles/s41587-022-01284-4>

We thank the reviewer for sharing this paper. The paper proposes a method (GLUE) for the analysis of multi-modal (non-spatial) single-cell data and uses adversarial learning to align the latent space of the different modalities while retaining the regulatory interactions across the multiple modalities through a graph variational autoencoder. The graph uses genes, ATAC peaks, or other modality-specific features as nodes. Edges between nodes are connected based on prior knowledge of interactions. The input modalities are decoded from the inner product of the graph latent representation and the latent space of the modality-specific encoders. This method is completely different to STACI:

1. GLUE, the paper cited by the reviewer, deals with single-cell expression data, while our manuscript is aimed at integrating spatial transcriptomics data with chromatin imaging that could provide major insights into biological processes in the tissue microenvironment going beyond single-cell analysis. In addition, the use of graph neural networks is fundamentally different: in our manuscript the graph encodes spatial relationships of cells, while in GLUE it encodes gene regulatory relationships.
2. GLUE cannot be used to integrate images or other modalities where no known regulatory information is available. To be more precise, images cannot directly be represented as a list of features similar to genes in sequencing-based data, which is required as the nodes of the graphs used by GLUE. Even when using an autoencoder to extract image features in the latent space, the latent features cannot be connected to other sequencing modalities through graphs, since the regulatory interactions between genes and latent image features are unknown. STACI is able to integrate imaging data with sequencing data by directly aligning the latent space of the modality-specific autoencoders, without relying on prior knowledge. With respect to the applications considered by GLUE, STACI could be helpful in cases where current knowledge of regulatory relationships is incomplete or when regulatory interactions are context-dependent. In contrast to GLUE, STACI is not dependent on the accuracy or completeness of prior knowledge of regulatory interactions.
3. In addition to learning a joint latent space for downstream analysis, such as clustering, STACI is also capable of predicting one modality from another input modality. We demonstrated such translation between modalities by predicting gene expression from chromatin images. GLUE has not been shown to be able to translate between modalities.

Of course, graph-based approaches as well as autoencoder-based approaches are being used widely for high-throughput data analysis. To provide just one other recent example, a supervised graph-based neural network method has been developed to infer ligand-receptor interactions from spatial transcriptomics data.¹¹ However, STACI is unique in its ability to train one model for integrating all modalities with built-in batch correction to learn a joint latent representation of all modalities including the spatial location of cells, which, as we demonstrated in the context of Alzheimer's, can be used to identify biologically meaningful tissue regions and enables the translation between different data modalities.

[How to balance the reconstruction errors from each modalities]

I was thinking about what if you transform the gene data or normalize/scale them in a different way, will this result a dramatical change in joint training?

We thank the reviewer for this question. Min-max scaling has been used as the standard normalization technique in neural networks, especially in convolutional networks¹². In all our works, we have found this normalization to work well for gene expression data and lead to stable training converging to low values of the loss function. For example, the gene expression reconstruction error is much higher when using the standard normalization used in statistical (by which we here mean methods that don't make use of neural networks) gene expression analysis methods, i.e. library-size normalization, log transformation, and z-score normalization of each gene (in Figure 3 below compare the 3rd row to the 1st and 2nd rows). In terms of balancing the reconstruction errors from each modality, the reconstruction error of gene expression is the same in the autoencoder with only gene expression (see below in Figure 3, 1st row) compared to the full STACI model with both gene expression and cell adjacency (see below in Figure 3, 2nd row); all results are shown on the 10% held-out cells. Thus, our graph-based autoencoder still achieves the minimum test loss of gene expression reconstruction even when an additional modality is incorporated and is thus robust to balancing the different reconstruction errors.

test loss	8-month control	8-month disease	13-month control	13-month disease
gene expression only	0.06849	0.063415	0.067315	0.072015
log transform + min-max scaling	0.06863	0.063495	0.06747	0.072105
library-size normalization, log transform, z-score normalization	0.1486	0.13604	0.143135	0.155555

Figure 3. Gene expression reconstruction errors of STACI on the 10% held-out cells. Here, 'gene expression only' refers to the default log transformation, min-max scaling, and training the model using only gene expression without using cell adjacency; 'log transform + min-max scaling' refers to the full STACI model with the default normalization and using both gene expression and cell adjacency as the inputs and outputs; row 3 refers to the same architecture as in row 2 but the gene expression input is normalized by library-size normalization, log-transformation, and z-score normalization of the genes.

[Code]

I check the github and saw experiments with "convolution exploded"

(https://github.com/uhrerlab/STACI/blob/master/train_jointGAEcnnRegrs_starmap_multisamplesMixed.ipynb)

We thank the reviewer for pointing out this confusing output of our code. This particular notebook was not used in our manuscript and is part of an earlier version of STACI when we were still developing it. It was included by mistake and we thank the reviewer for noticing this. In contrast to the final STACI model, in this earlier version, we attempted to train the regression of plaque size simultaneously with the training of the joint latent space, which made the training less stable. In the final version, the model for the joint latent space and the model for the plaque size regression were trained separately. The final version of STACI shows very stable training:

https://github.com/uhrerlab/STACI/blob/master/train_jointGAEcnn_starmap_multisamples.ipynb.

"name errors"

(https://github.com/uhrerlab/STACI/blob/master/train_jointGAEcnn_starmap_multisamples.ipynb)

We thank the reviewer for pointing out this error message. The training of our model completed in cell 9 of the notebook (where all the losses are printed out). Similarly as in the comment above, this part of the notebook was just used for testing purposes during the development of our method and was not used in the manuscript. We removed it from the notebook to avoid confusion and thank the reviewer for noticing it.

This is not something I expect from github repository. I was thinking can you provide a demonstration for readers showing how to use your tool?

We have extended the README file as well as the notebooks to provide more detailed instructions of how to run our code. A user just needs to specify the paths to where their data is stored and where to save the outputs before running each cell of the notebooks in order. Hyperparameter changes, such as changing the number of latent dimensions, can also be done in the notebooks according to the instructions provided. A screenshot of the notebook for training the graph autoencoder with the added instructions is shown in Figure 4 below.

```

In [3]: 1 # Settings
2 os.environ["CUDA_VISIBLE_DEVICES"] = "2" #this should be set to the GPU device you would like to use on y
3 use_cuda=True #set to true if GPU is used
4 fastMode=False #Perform validation during training pass
5 seed=3 #random seed
6 useSavedMaskedEdges=True #some edges of the adjacency matrices are held-out for validation; set to True to
7 maskedEdgeName='knn20_connectivity'
8 epochs=10000 #number of training epochs
9 saveFreq=30 #the model parameters will be saved during training at a frequency defined by this parameter
10 lr=0.001 #initial learning rate
11 lr_adv=0.001 #this is ignored if not using an adversarial loss in the latent space (i.e. it is ignored for
12 weight_decay=0 #regularization term
13
14 hidden1=5000 #Number of units in hidden layer 1
15 hidden2=5000 #Number of units in hidden layer 2
16 # hidden3=2048 # dimensions of additional hidden layers in the encoder, if more layers are specified
17 # hidden4=2048
18 # hidden5=128
19 fc_dim1=5000 #Number of units in the fully connected layer of the decoder
20 # fc_dim2=128 # dimensions of additional hidden layers in the decoder, if more layers are specified
21 # fc_dim3=128
22 # fc_dim4=128
23 adv_hidden=128 #ignored if not using an adversarial loss in the latent space. This is the hidden units of
24
25 dropout=0.01 #neural network dropout term
26 testNodes=0.1 #fraction of total cells used for testing
27 valNodes=0.05 #fraction of total cells used for validation
28 XreconWeight=20 #reconstruction weight of the gene expression
29 advWeight=2 # weight of the adversarial loss, if used
30 model_str='gcn_vae_xa_e2_d1_dca' #specify which model to use (see definition below): 'gcn_vae_xa_e2_d1_dca
31 adv=None # different choices of the adversarial loss, if used (as defined below): 'clf_fc1_eq', 'clf_fc1
32 ridgel=0.01 #regularization weight of the gene dropout parameter
33 shareGenePi=True #ignored in the default model; This is a parameter to specify how if the gene dropout ter
34
35 num_features=2112 #number of input genes
36 training_samples=['control13','disease13','control8','disease8'] #names of the input samples used for tra
37 targetBatch=None #if adversarial loss is used, one possibility is to make all batches look like one target
38 training_sample_X='logminmax' #specify the normalization method for the gene expression input. 'logminmax'
39 switchFreq=10 #the number of epochs spent on training the model using one sample, before switching to the
40 standardizeX=False #if perform additional z-score normalization of genes. Default is False.
41 name='allk20XA_05_dca_over_lossXreconOnly_wKL' #name of the model
42 useA=False #set to True to include adjacency loss as in the full STACI model
43
44 #provide the paths to save the training log, trained models, and plots, and the path to the directory whe
45 logsavepath='/mnt/external_ssd/xinyi/log/train_gae_starmap/'+name
46 modelsavepath='/mnt/external_ssd/xinyi/models/train_gae_starmap/'+name
47 plotsavepath='/mnt/external_ssd/xinyi/plots/train_gae_starmap/'+name
48 datadir='/home/xinyiz/2021-01-13-mAD-test-dataset'
49
50 #Load data
51 sampleidx={'disease13':'AD_mouse9494',
52            'control13':'AD_mouse9498',
53            'disease8':'AD_mouse9723',
54            'control8':'AD_mouse9735'} #this is formatted as {name of the sample as used in 'training_sample
55 savedir=os.path.join('/home/xinyiz/starmap') #where pre-computed adjacency matrices are stored
56 adj_dir=os.path.join(savedir,'a')
57
58 #normalize the gene expression or load the normalized gene expression from Hu et al.
59 #batch information should be stored in the metadata as 'sample'
60 featureslist={}
61 if training_sample_X in ['corrected','scaled']:
62     scaleddata=scanpy.read_h5ad(datadir+'/2020-12-27-starmap-mAD-scaled.h5ad') #change to the h5ad file ne
63

```

Figure 4. A screenshot of the updated notebook with more detailed instructions.

Reference

1. Multiomic Integration Neuroscience Application Note: Visium for FFPE Plus Immunofluorescence Alzheimer's Disease Mouse Model Brain Coronal Sections from One Hemisphere Over a Time Course. *10x Genomics*
<https://www.10xgenomics.com/resources/datasets/multiomic-integration-neuroscience-application-note-visium-for-ffpe-plus-immunofluorescence-alzheimers-disease-mouse-model-brain-coronal-sections-from-one-hemisphere-over-a-time-course-1-standard>.
2. Haghverdi, L., Lun, A. T. L., Morgan, M. D. & Marioni, J. C. Batch effects in single-cell RNA sequencing data are corrected by matching mutual nearest neighbours. *Nat. Biotechnol.* **36**, 421–427 (2018).
3. Leek, J. T. & Storey, J. D. Capturing Heterogeneity in Gene Expression Studies by Surrogate Variable Analysis. *PLOS Genet.* **3**, e161 (2007).
4. Johnson, W. E., Li, C. & Rabinovic, A. Adjusting batch effects in microarray expression data using empirical Bayes methods. *Biostatistics* **8**, 118–127 (2007).
5. Eraslan, G., Simon, L. M., Mircea, M., Mueller, N. S. & Theis, F. J. Single-cell RNA-seq denoising using a deep count autoencoder. *Nat. Commun.* **10**, 390 (2019).
6. Zeng, H. *et al.* Integrative in situ mapping of single-cell transcriptional states and tissue histopathology in an Alzheimer's disease model. *bioRxiv* (2022).
7. Pham, D. *et al.* stLearn: integrating spatial location, tissue morphology and gene expression to find cell types, cell-cell interactions and spatial trajectories within undissociated tissues. 2020.05.31.125658 Preprint at <https://doi.org/10.1101/2020.05.31.125658> (2020).
8. Zhu, Q., Shah, S., Dries, R., Cai, L. & Yuan, G.-C. Identification of spatially associated subpopulations by combining scRNAseq and sequential fluorescence in situ hybridization data. *Nat. Biotechnol.* **36**, 1183–1190 (2018).
9. Bao, F. *et al.* Integrative spatial analysis of cell morphologies and transcriptional states with MUSE. *Nat. Biotechnol.* 1–10 (2022) doi:10.1038/s41587-022-01251-z.
10. Lein, E. S. *et al.* Genome-wide atlas of gene expression in the adult mouse brain. *Nature* **445**, 168–176 (2007).
11. Yuan, Y. & Bar-Joseph, Z. GCNG: graph convolutional networks for inferring gene interaction from spatial transcriptomics data. *Genome Biol.* **21**, 300 (2020).
12. Géron, A. *Hands-on Machine Learning with Scikit-Learn, Keras, and TensorFlow: Concepts, Tools, and Techniques to Build Intelligent Systems.* (O'Reilly Media, Incorporated, 2019).

Reviewers' Comments:

Reviewer #3:

Remarks to the Author:

The authors have addressed all of my concerns.